



**Measurement report: Changes in light absorption and molecular**
**composition of water-soluble humic-like substances during a winter**
**haze bloom-decay process in Guangzhou, China**
Chunlin Zou[1,3], Tao Cao[1,3], Meiju Li[1,3], Jianzhong Song[1,2,4,*], Bin Jiang[1,2], Wanglu Jia[1,2],
Jun Li[1,2], Xiang Ding[1,2], Zhiqiang Yu[1,2,4], Gan Zhang[1,2], Ping'an Peng[1,2,3,4]
[1]State Key Laboratory of Organic Geochemistry and Guangdong Provincial Key
Laboratory of Environmental Protection and Resources Utilization, Guangzhou Institute
of Geochemistry, Chinese Academy of Sciences, Guangzhou 510640, China
[2]CAS Center for Excellence in Deep Earth Science, Guangzhou 510640, China
[3]University of Chinese Academy of Sciences, Beijing 100049, China
[4]Guangdong-Hong Kong-Macao Joint Laboratory for Environmental Pollution and
Control, Guangzhou 510640, China
*Correspondence to*: Jianzhong Song (songjzh@gig.ac.cn)





**Abstract**
Water-soluble humic-like substances (HULIS) absorb light in near-UV and visible
wavelengths and exert significant influence on the atmospheric environment and climate.
However, knowledge on HULIS evolution during haze bloom-decay process is limited.
Herein, $PM_{2.5}$ samples were obtained during a winter haze event in Guangzhou, China,
and light absorption and molecular composition of HULIS were investigated by UV-vis
spectrophotometry and ultrahigh-resolution mass spectrometry. Compared with HULIS
in clean days, the absorption coefficients ($Abs_{365}$) of HULIS in haze days were
significantly higher but the mass absorption efficiencies ($MAE_{365}$) were relatively lower,
suggesting diverse and dynamic absorption properties of HULIS during haze episodes.
The CHO and CHON compounds were the most abundant components in HULIS,
followed by CHOS, CHONS, and CHN. Haze HULIS presented comparatively higher
molecular weight, lower aromaticity index ($AI_{mod}$), and higher $O/C_w$, $O/N_w$, and $O/S_w$
ratios, indicating that HULIS fractions undergo relatively higher oxidation during haze
days than clean days. Moreover, CHON and CHO compounds with high $AI_{mod}$ were the
major potential chromophores in HULIS and significantly contributed to HULIS light
absorption. It's worth noting that the proportions of these chromophores were decreased
during haze event, mainly owing to their higher oxidation and longer aging period during
haze episode. Besides, accumulated contribution of organic compounds emitted from
vehicles and formed from stronger reactions of bio-VOCs also diluted light-absorbing
compounds in haze HULIS. These findings help to understand HULIS evolution during
haze bloom-decay process in the subtropic region of China.



**1. Introduction**

Water-soluble humic-like substances (HULIS), belonging to a class of highly complex organic compounds with physical/chemical properties similar to humic substances in natural environments, which constitute 30%−70% of water-soluble organic compounds in ambient aerosols and are responsible for > 70% of light absorption in water-soluble brown carbon (BrC) (Graber and Rudich, 2006; Laskin et al., 2015; Huang et al., 2018). HULIS are ubiquitously identified in atmospheric aerosols, fog, cloud, and rain water, and have been demonstrated to play significant effects on both atmospheric environment and climate (Bianco et al., 2018; Wu et al., 2018b; Zeng et al., 2021). In addition, HULIS exert adverse health effects because they can enhance the oxidative potential of organic aerosols (Ma et al., 2019; Wong et al., 2019; Chen et al., 2019).

In recent years, severe particulate pollution (i.e., haze events) frequently occur in some developing country such as China, which has drawn extensive public and scientific concerns (Huang et al., 2014; Wang et al., 2014; An et al., 2019; Zhang et al., 2020). According to An et al. (2019), contributions of organic aerosols, including primary organic aerosols and secondary organic aerosols (SOA), are significant for severe haze events; in particular, the contribution of SOA in China is expected to continuously increase because of stronger chemical reactions in the atmosphere (An et al., 2019). HULIS are an important component in organic aerosols, which originate from a variety of primary emissions (e.g., biomass burning (BB), coal combustion, off-road engine emission) (Fan et al., 2016; Cui et al., 2019; Tang et al., 2020) and secondary chemical oxidation of biogenic and anthropogenic volatile organic compounds (VOCs) (Rincón et al., 2009; Tomaz et al., 2018) and soot (Fan et al., 2020). During the haze episode, a



number of chemical processes occur in aqueous phase (Wong et al., 2017, 2019; Wu et
al., 2018b) and gas phase (Sumlin et al., 2017), which lead to significant changes in
chemical composition and light absorption properties of HULIS. For instance, recent
studies on oxidation of BB-derived BrC have indicated that although both enhancement
and bleaching of BrC occur during aging, bleaching of BrC becomes dominant over a
long period (Fan et al., 2020; Wong et al., 2017, 2019; Ni et al., 2021). However,
multiphase reaction between carbonyl and amine has demonstrated rapid formation of
light-absorbing organic compounds (Kampf et al., 2016). Nevertheless, it should be noted
that these results were mainly obtained from laboratory experiments and may not reflect
the complex evolution behavior of BrC in atmospheric environment.

High concentrations of HULIS have been determined during typical haze episodes

in northern, eastern, and southern China (Song et al., 2016; Win et al., 2020; Zhang et al.,
2020; Wang et al., 2020), and have been demonstrated to significantly influence
atmospheric visibility, environment, and photochemical process. Guangzhou is the
biggest city in the Pearl River Delta (PRD), one of the most developed regions in China,
and is located in the subtropical zone with a population of over 18 million people (Yu et
al., 2017). Although a remarkable decline in atmospheric particulate matter ($PM_{2.5}$)
pollution has been observed in recent years owing to strict regulatory controls, $O_3$ and
VOCs still remain at higher levels and severe haze pollution caused by fine particulate
matter frequently occur in winter (Huang et al., 2014). Several studies have investigated
the chemical and optical properties of HULIS in the PRD region and found that both BB
and SOA formation have significant effects on these organic compounds in atmosphere
(Fan et al., 2016; Zhang et al., 2021; Jiang et al., 2020, 2021). However, detailed



information regarding the evolution of light absorption and molecular composition of
HULIS during haze events is still scarce.

Recently, ultrahigh-resolution Fourier transform ion cyclotron resonance mass

spectrometry (FT-ICR MS) coupled with electrospray ionization (ESI) sources has been
frequently employed to investigate the exact molecular characteristics of HULIS in
ambient aerosols (Tang et al., 2020; Song et al., 2018, 2022; Zeng et al., 2021). Owing to
its extremely high mass resolution and accuracy, this technique allows further exploration
of the evolution of HULIS during haze event. The present study performed
comprehensive characterization of HULIS in $PM_{2.5}$ collected during a haze event in
Guangzhou, China. The abundances and light absorption properties of HULIS were first
measured, and carbonaceous fractions, water-soluble ions, and levoglucosan (Lev) were
determined. Subsequently, four HULIS samples collected during different haze stages
were analyzed using FT-ICR MS operated in both ESI− and ESI+ modes. To our
knowledge, the present study is aim to apply a combination of optical properties and
molecular characterization by FT-ICR MS to investigate HULIS in a haze event in the
subtropical zone of China. The results obtained provide novel insights into the evolution
of HULIS during haze event, and are important for predicting the environmental and
climatic effects of HULIS in South China.
**2. Material and Methods**
**2.1. Aerosol sampling**
The $PM_{2.5}$ samples were collected on the campus of Guangzhou Institute of
Geochemistry, Chinese Academy of Sciences, Guangzhou, China (23.14N, 113.35E),



which is an academic and residential region. Traffic emissions and residential activities
are the potential pollution sources in the sampling area. The 24-h $PM_{2.5}$ sampling was
conducted using a high-volume sampler (Tianhong Intelligent Instrument Plant, Wuhan,
China, with a flow rate of 1.0 $m^3\,min^{-1}$) during 7 to 30 January of 2018, and a total of 24
samples were collected on the prebaked quartz filters ($20.3 \times 25.4\ cm^2$, Whatman,
Maidstone, UK). Field blank samples were collected without power on. After collection,
all filter samples were stored in a refrigerator at −20 °C until analysis. Meteorological
data (http://www.wunderground.com/history/airport/ZGGG), including wind speed,
temperature, relative humidity, and concentrations of $SO_2$, $O_3$, and $NO_2$, for the sampling
days are presented in Figure 1 and Table S1.
**2.2. Isolation of HULIS**

HULIS were isolated using a solid-phase extraction (SPE) procedure as described

previously (Zou et al., 2020). Briefly, portions of the $PM_{2.5}$ samples (100 $cm^2$) were
ultrasonically extracted with 50 mL of ultrapure water for 30 min. The extracts were
filtered through a 0.22-μm PTFE syringe filter and then adjusted to pH of 2 with HCl,
and loaded on a preconditioned SPE cartridge (Oasis HLB, 200 mg/6 mL, Waters, USA).
The hydrophilic fraction (i.e., inorganic ions, high-polar organic acids, etc) was removed
with ultrapure water, whereas the relatively hydrophobic HULIS fraction was retained
and eluted with 2% (v/v) ammonia/methanol. Finally, HULIS solution was evaporated to
dryness with a gentle $N_2$ stream and redissolved with ultrapure water for the analysis.
**2.3. Light absorption analysis**





The absorption spectra of the water-soluble organic carbon (WSOC) and HULIS
fractions were measured by a UV-vis spectrophotometer (UV-2600, Shimadzu) between
200 to 700 nm. Each spectrum was corrected for the filter blanks. The light absorption
coefficients, absorption Ångström exponent (AAE) and mass absorption efficiency
(MAE$_\lambda$) were calculated and the detailed methods are presented in the Supporting
Information (SI).
**2.4. FT-ICR MS analysis**
For FT-ICR MS analysis, the HULIS samples were isolated from PM$_{2.5}$ collected
during four periods: before haze days (clean-I days, 7−12 January), haze bloom days
(haze-I days, 13−18 January), haze decay days (haze-II days, 19−24 January), and after
haze days (clean-II days, 25−30 January). A filter punch (18 cm in diameter) was taken
from every sample, and all the six samples in each period was combined for the isolation
of HULIS fractions. The obtained HULIS samples were measured with an ESI FT-ICR
MS (Bruker Daltonik GmbH, Bremen, Germany) equipped with a 9.4 T refrigerated
actively shielded superconducting magnet. The system was operated in both ESI− and
ESI+ modes. The scan range was set to m/z from 100 to 1000, with a typical mass-
resolving power >450,000 at m/z 319 with <0.2 ppm absolute mass error. The mass
spectra were calibrated externally with arginine clusters and internally recalibrated with
typical O$_5$-class species peaks in DataAnalysis 4.4 (Bruker Daltonics). Due to the
inherent differences in the ionization mechanisms between ESI- and ESI+ modes, the
data detected by the two ionization modes can provide complementary information on the
molecular composition of atmospheric HULIS (Lin et al., 2012; Lin et al., 2018). The
details of data analysis are provided in the SI.





**2.5. Chemical analysis**
The amounts of organic carbon (OC) and elemental carbon (EC) were determined by
a OC/EC analyzer (Sunset Laboratory Inc., USA) (Mo et al., 2018). The concentrations
of WSOC and HULIS were determined by a TOC analyzer (Shimadzu TOC_VCPH,
Kyoto, Japan). The water-soluble inorganic species ($NO_3^-$, $SO_4^{2-}$, $Cl^-$, $NH_4^+$, $K^+$, $Na^+$,
$Ca^{2+}$, $Mg^{2+}$,) were measured with a Dionex ICS-900 ion chromatography system (Thermo
Fisher Scientific, USA) as described previously (Huang et al., 2018). The concentrations
of Lev were analyzed with a gas chromatography–MS after derivatization with BSTFA
and pyridine at 70 ℃ for 3 h (Huang et al., 2018). Detailed information regarding these
measurements is provided in the SI.
**3. Results and Discussion**
**3.1. Abundance and chemical composition of $PM_{2.5}$**
Figure 1 shows the meteorological conditions, $PM_{2.5}$ concentration, and
concentrations of major chemical constituents, including carbon fractions and water-
soluble inorganic ions in $PM_{2.5}$ samples obtained during a haze bloom-decay process.
Based on the variation in $PM_{2.5}$ concentration, these samples were categorized into four
groups: clean-I days (before haze, 14−24 μg m$^{-3}$), haze-I days (haze bloom, 45−114 μg
m$^{-3}$), haze-II days (haze decay, 58−115 μg m$^{-3}$), and clean-II days (after haze, 9−35 μg
m$^{-3}$). As indicated in Table S1 and Figure 1, the $PM_{2.5}$ concentrations increased from 18
± 3.3 μg m$^{-3}$ in clean-I days to 82 ± 26 and 84 ± 22 μg m$^{-3}$ in haze-I and haze-II days,
respectively, and then decreased to 21 ± 10 μg m$^{-3}$ in clean-II days. This finding
obviously indicated that the average $PM_{2.5}$ concentrations during the examined haze





episode are higher than the second-grade national ambient air quality standard in China

(75 µg m$^{-3}$, 24 h), whereas those during clean days are lower than the first-grade national

ambient air quality standard in China (35 µg m$^{-3}$, 24 h). However, the average PM$_{2.5}$

concentrations during the haze event are lower than those in the cities in winter haze,

including Shenyang (108 µg m$^{-3}$) (Zhang et al., 2020), and Nanjing (123 ± 28.5 µg m$^{-3}$)

(Li et al., 2020), Beijing (158 µg m$^{-3}$), and Xi'an (345 µg m$^{-3}$) (Zhang et al., 2018). As

shown in Figure 1a, the wind speed decreased from 4 to 1.5 m s$^{-1}$ during the haze bloom

process, resulting in stable meteorological conditions. Moreover, the PM$_{2.5}$ concentrations

were significantly negatively correlated with wind speed (r = −0.77; p < 0.01), indicating

that poor air dispersion condition leads to accumulation of particulate matter in the study

region.

As shown in Table S1, the average concentrations of OC and EC were 2.2−15 and

0.36−2.7 µgC m$^{-3}$ in the four stages, respectively, implying that the distinct changes in

OC and EC were higher during haze episodes than those in clear days. During the entire

study period, WSOC concentration ranged from 0.5 to 12.5 µgC m$^{-3}$ (4.3 ± 1.2 µgC m$^{-3}$),

which contributed to 53%−57% of OC in PM$_{2.5}$. The HULIS concentration noted in the

present study ranged from 0.15 to 6.1 µgC m$^{-3}$ (2.2 ± 1.9 µgC m$^{-3}$), which was

comparable to those observed in the PRD region, such as Hong Kong (2.38 ± 1.62 µgC

m$^{-3}$) (Ma et al., 2019) , Guangzhou (2.4 ± 1.6 µgC m$^{-3}$) (Fan et al., 2016), and Heshan

(2.08 ± 1.16 µgC m$^{-3}$) (Jiang et al., 2020), but lower than those  in northern cities of

China, such as Xi'an ( 12.4 ± 6.5 µgC m$^{-3}$) (Huang et al., 2020), Beijing (3.79 ± 3.03

µgC m$^{-3}$) (Mo et al., 2018), and Lanzhou (4.7  µgC m$^{-3}$) (Tan et al., 2016). As shown in

Figure 1, HULIS fraction also exhibited obvious variations during the entire sampling





period. The average HULIS concentration was $0.46 \pm 0.22$ µgC m$^{-3}$ in clean-I days,
which sharply increased to $4.5 \pm 1.2$ µgC m$^{-3}$ in haze-I days, then decreased to $3.1 \pm 1.2$
µgC m$^{-3}$ in haze-II days, and rapidly declined to $0.75 \pm 0.52$ µgC m$^{-3}$ in clean-II days.
This result was consistent with the changing trend of WSOC, OC, and EC. In addition,
the HULIS/WSOC ratios were about $0.50 \pm 0.13$ in the four PM$_{2.5}$ samples, which are in
broad agreement with other studies showing that HULIS is the major fraction of WSOC
(Fan et al., 2016; Ma et al., 2019; Jiang et al., 2020).
As illustrated in Figure 1, obvious variations in chemical compositions were also
observed in these PM$_{2.5}$ samples. Secondary inorganic aerosols (SIA) (i.e., SO$_4^{2-}$, NO$_3^-$,
and NH$_4^+$), OC, and EC exhibited a similar variation during the entire study period, and
their contents sharply increased from 10 January in clean-I days to 13−18 January in
haze-I days, then slowly decreased in haze-II days, and finally reached lower levels in
clean-II days. It must be noted that the increasing rate of EC was similar to that of SIA in
haze-I days, indicating that direct emissions and atmospheric reactions may play similar
roles in PM$_{2.5}$ increase during this haze bloom period. As indicated in Figure 1f, the
highest values of NO$_3^-$/SO$_4^{2-}$ were observed in haze-I days, implying the important
influence of traffic exhausts in the haze bloom period (Mo et al., 2018). In addition, the
high NO$_2$ and O$_3$ concentrations and the stable meteorological condition with high
temperature also led to the outburst of fine particulate pollution in this period. During
haze-II days, the SIA and OM contents in PM$_{2.5}$ slowly decreased, whereas the
concentrations of Na$^+$, Cl$^-$, and unidentified materials as well as the Lev/OC ratio in
PM$_{2.5}$ increased (Figure 1e,f,h), suggesting that local contribution weakened and regional
contribution via BB and sea salt became more important(Jiang et al., 2021). This





phenomenon was also observed to be consistent with the changes in the pollutant sources
transported by air masses. As indicated in Figure S1, the $PM_{2.5}$ samples in haze-II days
included some contributors transported from coastal area of eastern Guangdong Province
and Fujian Province, and the $PM_{2.5}$ particulates are likely to be enriched with sea salt
materials and mineral dusts.

**3.2. Light absorption**

The light absorption properties of WSOC and HULIS (Figure 1d, i, j and Table S2)
exhibited obvious temporal variations during the sampling period. The AAE values of
WSOC and HULIS ranged from 4.1 to 6.4 and 5.6 to 6.6, respectively. The AAE values
for HULIS were obviously higher than those for WSOC in the same sample (Figure 1i),
indicating that light absorption of HULIS is more wavelength-dependent than that of
WSOC. This difference may be related with the higher enrichment of light-absorbing
organic species in HULIS. Moreover, the AAE values of HULIS did not present
significant variation during the entire haze process.
Light absorption at 365 nm ($Abs_{365}$) for WSOC and HULIS were $2.5 \pm 2.0$ and $1.8 \pm$
$1.6$ M m$^{-1}$, respectively (Table S2). HULIS contributed to about 72% of light absorption
coefficients by WSOC, implying that they enriched the major light-absorbing
components in WSOC. As shown in Figure 1d, the $Abs_{365}$ values for HULIS presented
obvious temporal variations. The $Abs_{365,HULIS}$ value was $0.55 \pm 0.06$ M m$^{-1}$ in clean-I
days, which first increased to $3.4 \pm 1.5$ M m$^{-1}$ in haze-I days and then slowly decreased to
$2.6 \pm 0.85$ M m$^{-1}$ in haze-II days, and finally rapidly declined to $0.64 \pm 0.32$ M m$^{-1}$ in
clean-II days. This result was similar to the variations in the mass concentration of
HULIS. Furthermore, the $Abs_{365}$ values for HULIS in Guangzhou were found to be





higher than those observed in southeastern Tibetan Plateau (0.38–1.0 M m$^{-1}$) (Zhu et al.,
2018), but obviously lower than those in Xi'an (7.6–36 M m$^{-1}$) (Shen et al., 2017) and
Beijing, (3.7–10.1 M m$^{-1}$) (Du et al., 2014).

In general, MAE$_{365}$ value can be used to assess the light absorption capacity of target

organic compounds (Li et al., 2019). As shown in Figure 1j and Table S2, the MAE$_{365}$
values for WSOC and HULIS were in the range of 0.68–1.3 m$^2$ gC$^{-1}$ (1.0 ±0.21 m$^2$ gC$^{-1}$)
and 0.77–1.8 m$^2$ gC$^{-1}$ (1.1±0.27 m$^2$ gC$^{-1}$), respectively, during the entire sampling period.
The generally higher MAE$_{365}$ values for HULIS, when compared with those for WSOC
demonstrated that HULIS are enriched with strong light-absorbing compounds. Moreover,
the MAE$_{365}$ values for HULIS measured in the present study were noted to be
comparable to those determined in Beijing (1.43 ± 0.33 m$^2$ g C$^{-1}$) (Mo et al., 2018),
Xi'an (0.91–1.85m$^2$ g C$^{-1}$) (Yuan et al., 2021), and Hong Kong (1.84 ± 0.77 m$^2$ gC$^{-1}$)
(Ma et al., 2019). The average MAE$_{365}$ values for HULIS exhibited some temporal
variations. The MAE$_{365}$ values for HULIS were 0.91±0.03 and 0.95 ± 0.11 m$^2$ gC$^{-1}$ in
haze-I and haze-II days, respectively, which were much lower than those (1.3 ± 0.22 and
1.3 ± 0.27 m$^2$ gC$^{-1}$, respectively) observed in clean-I and clean-II days, suggesting that
HULIS have a relatively weaker light absorption capability in haze days. This finding is
consistent with the results reported by Zhang et al. (2017), who found that the MAE$_{365}$
values in the heating or non-heating seasons during hazy days were lower than those in
clean days. These differences in MAE$_{365}$ values may potentially contribute to the stagnant
conditions prolonging secondary oxidation reaction or photolytic aging period, which
help the chromophores containing C=C unsaturated bond to be severely oxidized (Wang
et al., 2017a; Zhang et al., 2017). This process of reducing light absorption is also known



as "photobleaching" or photolytic aging, which often occurs in the aging process
(Forrister et al., 2015; Wong et al., 2017). Besides, an increase in additional sources for
HULIS in the study area, such as weaker or non-light-absorbing compounds formed by
atmospheric oxidation, could also result in weaker light absorption of HULIS during the
haze episode (Liu et al., 2018).

**3.3. Molecular evolution of HULIS during the haze process**

For an in-depth understanding of the variation in HULIS at molecular level during

the haze process, the four HULIS samples collected in different stages of the haze
process were analyzed by ESI FT-ICR MS in both negative and positive modes. As
shown in Figure 2, thousands of peaks were detected in the mass range between m/z 100
and m/z 700, with the high intensity ions noted within m/z 150–400 (Figure 2), which are
comparable to those of ambient HULIS determined in previous studies (Song et al., 2018,
2019, 2022; Wang et al., 2017b; Mo et al., 2018). As shown in Figure 2, some organic
compounds with stronger arbitrary abundance were labeled, and their formulas, double
bond equivalent (DBE), modified aromaticity index ($AI_{mod}$), and potential sources were
listed in Table S3. Compounds a ($C_7H_7NO_3$) and b ($C_8H_6O_4$), both have high DBE values,
which might be assigned to aromatics such as methylnitrophenol and phthalic acid.,
whereas compound d ($C_8H_{18}O_4S$) with low DBE value and high O/S ratio was probably
aliphatic organosulfate. These results suggested that both BB and vehicular emissions are
important sources of BrC in ambient aerosols (Mohr et al., 2013; Riva et al., 2015; Lin et
al., 2012). Furthermore, compound e ($C_{10}H_{17}NO_7S$) and compound f ($C_{10}H_{18}N_2O_{11}S$) in
Table S3 were found to be identical to the oxidation products of monoterpenes, suggest
that biogenic sources could contribute to the formation of HULIS (Surratt et al., 2008;



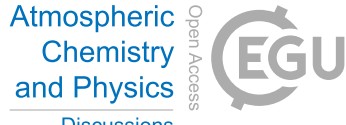

Wang et al., 2019b). Thus, HULIS could be affected by multiple sources during the haze
process, possibly including BB, biogenic sources, and anthropogenic emissions.

The identified formulas could be divided into seven compound categories, namely,

CHO−, CHON−, CHOS−, and CHONS− detected in ESI− mode and CHO+, CHN+, and
CHON+ detected in ESI+ mode. As illustrated in Figure 2, the CHO compounds were the
most abundant group in all the HULIS, accounting for 43%–50% and 51%–57% of the
overall compounds detected in the ESI− and ESI+ modes, respectively. It must be noted
that relatively lower contents of CHO− were detected during the haze episode (haze-I and
haze-II days) and CHO+ molecules in haze-I HULIS. The CHON compounds were the
second most abundant group in all the HULIS. As shown in Figure 2, the relative content
of CHON− was 23% in clean-I days, which slightly increased to 24%–25% in haze
episode, and then decreased to 23% in clean-II days. In contrast, the relative content of
CHON+ compounds was 41% in clean-I days, which increased to 45% in haze-I days,
then fell to 42% in haze-II days and 41% in clean-II days. Both CHOS− and CHONS−
compounds were identified in all the four HULIS, accounting for 19%–22% and 8%–11%
of the total identified compounds, respectively. The CHN+ compounds were the least
abundant (1.3%−3.6%) in the four HULIS samples, and were relatively higher during the
haze episode, especially in haze-I days.

Tables S4 and S5 show the relative abundance weighted elemental ratios, molecular

weight (MW), DBE, $AI_{mod}$, and carbon oxidation state ($OS_C$) for the identified
compounds in HULIS. The $MW_w$ values for HULIS determined in the ESI− mode in
haze-I and haze-II days were 302 and 283, respectively, which were higher than those in
clean-I and clean-II days (266 and 264, respectively). Similar variation was also observed





for $MW_w$ for HULIS detected in ESI+ mode (Table S5). These results clearly indicated
that higher MW compounds constituted HULIS obtained during the haze episode. It has
been reported that high MW BrC compounds have relatively higher chemical resistance
and are the long-lived components in atmospheric aerosols (Dasari et al., 2019; Wong et
al., 2017). Therefore, HULIS compounds may undergo longer aging and higher oxidation
during haze episode, and are thereby characterized by relatively high MW values.

Furthermore, the molecular properties of HULIS in different stages of haze process

also exhibited some observable differences. As shown in Table S4, the HULIS samples in
haze episode detected by ESI− mode presented relatively lower $AI_{mod,w}$ values and
relatively higher $O/C_w$, $O/N_w$, and $O/S_w$ ratios than those in clean days, indicating that
haze HULIS exhibited relatively lower aromaticity and higher oxidation degree than
clean HULIS. These differences can be attributed to bleaching or degradation of aromatic
compounds (i.e., nitroaromatic compounds or polycyclic aromatic hydrocarbons (PAHs))
by photooxidation during the haze process. In addition, increased contribution from
traffic emission and secondary reactions of bio-VOCs also decreased the aromaticity and
increased the oxidation degree of HULIS (Liu et al., 2016; Tang et al., 2020). These
changes in HULIS compounds led to the decrease in their $MAE_{365}$ values during the haze
episode, as described above ( Zhong and Jang, 2014; Song et al., 2019).

### 3.3.1. CHO Compounds

The CHO compounds bear O-containing functional groups, and have been

frequently detected in ambient aerosols. As shown in Figure 2, the CHO compounds were
the predominant component in the four HULIS samples, and the $MW_w$ values for CHO−
and CHO+ compounds were 247−288 and 236−272, respectively, with relatively higher





$MW_w$ values observed for the CHO group (CHO− and CHO+) in haze HULIS, especially
in haze-I samples. This finding is comparable to that reported in a previous study in
which aqueous oxidation of BB mixtures was found to yield high MW of organic
products (Tomaz et al., 2018; Yu et al., 2016).

The $OS_C$ is often used to describe the degree of oxidation of organic species in the

atmosphere (Kroll et al., 2011; Tong et al., 2019; Kourtchev et al., 2016). Figure 3 shows
plots of $OS_C$ versus carbon number for the CHO compounds. As indicated in the figure,
CHO compounds exhibited $OS_C$ from −2 to +1 with up to 40 carbon atoms. Kroll et al.
(2011) proposed that compounds with $OS_C$ between −0.5 and +1 and < 18 carbon atoms
can be attributed to semi-volatile and low-volatile oxidized organic aerosols (SV-OOA
and LV-OOA), which are mainly formed by complex oxidation reactions in atmosphere.
Compounds with $OS_C$ between −0.5 and −1.5 and 6–23 carbon atoms are related to
primary biomass burning organic aerosol (BBOA). In addition, compounds with $OS_C$
between −1 and −2 and ≥18 carbon atoms have been suggested to be hydrocarbon-like
organic aerosols (HOA), which are regarded as primary combustion surrogate (Zhang et
al., 2005; Kroll et al., 2011; Wang et al., 2017b).

As illustrated in Figure 3 and Table S6, most of the CHO− compounds clustered in

the BBOA region, accounting for 40%–46% of the total CHO− compounds, thus
suggesting that BB may be a major contributor to CHO compounds in HULIS. Figure 3
clearly indicates that the majority of aromatic and condensed aromatic compounds
produced signals in the $OS_C$ region between −0.5 and 1.0 and carbon number of 3–18
(Figure 3), which corresponded to SV-OOA and LV-OOA. The proportions of SV-OOA
and LV-OOA accounted for 23%–28% and 1.9%–2.4% of the total CHO− compounds,





respectively, and presented no significant variation. In contrast, the HOA components in
haze-I days showed the highest abundance (18%), which were much higher than those
(3.5%–4.5%) in haze-II, clean-I, and clean-II days. This finding indicated that the
increase in the primary source is associated with fossil fuel combustion such as vehicle
emissions during the haze bloom period (Zhang et al., 2005).

As shown in Figure 3, CHO+ compounds presented lower $OS_C$ (from −2.0 to 1.0)

than CHO− compounds. Most of the CHO+ compounds occurred in the BBOA region in
all four HULIS samples, making up to 60%–72% of the total CHO+ compounds, which
were much higher than those detected in ESI− mode, indicating that primary organic
compounds produced from BB were preferably detected in ESI+ mode. The HOA among
CHO+ compounds showed the same changing trends as those among CHO− compounds,
and higher HOA abundance was observed during haze-I days. In addition, some high
$AI_{mod}$ values of aromatics were found in the regions A1+ and A2+ (Figure 3), which
implied that the highest $AI_{mod}$ values (AI ≥ 0.67) with DBE ≥ 22 were only detected
during the haze days possibly owing to soot-derived materials or oxidized PAHs
(Decesari et al., 2002; Kuang and Shang, 2020). It must be noted that the sampling site in
the present study is influenced by traffic sources, causing increased accumulation of
vehicle-exhausted soot during haze episode, which was confirmed by relatively low
BBOA content in haze-I days. Oxidation of soot particles could result in the formation of
water-soluble high aromatic organic species (Decesari et al., 2002).
**3.3.2. CHON Compounds**

In the present study, 1379–2217 and 2008–2943 formulas were assigned to CHON

compounds identified in the ESI− and ESI+ spectra, respectively, which accounted for





23%–25% (ESI−) and 41%–45% (ESI+) of total identified compounds, respectively.
Relatively higher contents of CHON− compounds were obviously detected in HULIS
samples obtained during haze-I days, suggesting the occurrence of more N-containing
components in HULIS during haze bloom days. As shown in Tables S4 and S5, the
average $MW_w$ values for CHON− and CHON+ compounds were 328 and 317 in haze-I
days, respectively, which were slightly higher than those determined in haze-II days and
all higher than those observed in clean-I and clean-II days. Meanwhile, the $AI_{mod,w}$ values
for CHON− in haze days were 0.31–0.34, which were slightly lower than those in clean
days (0.37 and 0.40). These findings indicated that more high MW CHON compounds
with lower aromatic structures were formed during the haze episode.
The $O/N_w$ ratios for CHON− and CHON+ during haze-I and haze-II days were 5.3–
5.7 and 3.8, respectively, which were higher than those determined during the two clean
periods, confirming that these compounds were highly oxidized during the haze episode
(Tables S4 and S5). In general, compounds with $O/N \geq 3$ may indicate oxidized N groups
such as nitro ($-NO_2$) or nitrooxy ($-ONO_2$), whereas compounds with $O/N < 3$ may denote
the reduced N compounds (i.e., amines) (Lin et al., 2012; Song et al., 2018). In the
present study, most of the CHON compounds (79%–91% of CHON− compounds and
61%–64% of CHON+ compounds) exhibited $O/N \geq 3$, suggesting that high
concentrations of nitro compounds or organonitrates were contained in the CHON
compounds. Moreover, these compounds were more abundant in the CHON− group
during the haze episode (87%–91%), when compared with those during clean-I and
clean-II days (79%–82%), again implying that CHON− compounds undergo relatively
higher oxidization during the haze episode. As indicated in Figure 1, the increase in $NO_2$





was consistent with increased production of highly oxidized N-containing organic
compounds (NOCs) during the haze episode, which suggested the significant contribution
of $NO_3$-related multigenerational chemistry to organonitrate aerosol formation
(Berkemeier et al., 2016).
The majority of aromatics and condensed aromatics produced clear signals in
regions associated with SV-OOA and LV-OOA (Figure 4). BBOA also constituted a
significant proportion (33%−39%) in the CHON− group, and a relatively lower BBOA
content was observed in haze-I days. The abundance of HOA was relatively lower,
accounting for 2.3%−7.8% of the total CHON compounds, and the relative abundance of
HOA in haze-I days was much higher than that in haze-II, clean-I, and clean-II days,
suggesting the accumulation of primary fossil fuel combustion during haze-I days.
The CHON+ compounds mainly occurred at the range of $−2.0 < OS_C < 1.5$, with
average $OS_C$ values of around −1.0 for each sample, clearly indicating that CHON+
compounds were relatively lower than CHON− compounds. Most of the CHON+
compounds were detected in the BBOA region, accounting for 60%−76% of the total
$CHON^+$ compounds. The relative contribution of BBOA in haze-I days was lower than
that in haze-II and clean days. Moreover, a large number of aromatic species were
observed at the region B1+ (Figure 4), demonstrating that higher aromatic compounds
were only detected in haze-I days, which may be related to soot or BC. Similar trend was
also exhibited by CHO+ compounds, indicating the important role of local combustion
sources (e.g., traffic emission) during haze-I days.






### 3.3.3. CHOS and CHONS Compounds


In this study, 478–696 CHOS compounds and 306–589 CHONS compounds were
identified in ESI– mode (Table S4). Among these S-containing compounds, >86% of the
CHOS compounds had O/S ratios >4, whereas > 89% of the CHONS compounds
presented O/S ratios >7, suggesting that these S-containing compounds were possibly
organosulfates and nitrooxyorganosulfates. As listed in Table S4, the $AI_{mod,w}$ values for
CHOS and CHONS were about 0.02 and 0.01 in the HULIS fraction, which were much
lower than those for CHO and CHON. Almost 99% of the CHOS and CHONS
compounds in the HULIS fraction had $AI_{mod}$ values <0.5, while >93% of the CHONS
compounds had $AI_{mod} = 0$, indicating that they were mainly comprised of aliphatic and
olefinic organosulfates. These results are consistent with the previous findings that the
major S-containing compounds among organic aerosols in Guangzhou are organosulfates
formed by secondary oxidation reaction of long-chain alkenes/fatty acids with $SO_2$ (Jiang
et al., 2020), which generally possessed long aliphatic carbon chains and a higher degree
of oxidation. However, these compounds are different from the S-containing compounds
detected during the hazy days in Beijing (Jiang et al., 2016; Mo et al., 2016), which were
determined to be aliphatic organosulfates with low degree of oxidation and higher
amounts of aromatics and PAH-derived organosulfates, having a strong correlation with
anthropogenic emissions.
As described earlier, CHOS− and CHONS− compounds might be related to
organosulfates or nitrooxyorganosulfates, which have been observed to be derived from
atmospheric reactions of bio-VOCs such as α-pinene, limonene, and isoprene (Huang et
al., 2018; Surratt *et al.*, 2008) and fossil fuel combustion including coal combustion, off-





road engine emissions (Song et al., 2018, 2019; Cui et al., 2019). In the present study, the
relative contents of S-containing compounds (CHOS+CHONS) in the HULIS fraction in
haze days were all higher than those in clean days (Figure 2). Moreover, the CHOS and
CHONS compounds in haze HULIS always have relatively high relatively O/S ratios
than those in clean HULIS. These findings suggested the relatively higher contribution of
$SO_2$-related chemical oxidation during the haze event.
**3.3.4. CHN Compounds**
The N-bases (CHN) are usually identified in ambient aerosols and smokes from BB.
In the present study, 110–165 CHN+ compounds were identified in ESI+ mode, with
most of them (>86%) presenting DBE $\geq$ 2, suggesting that they might be nitrile and
amine species (Lin et al., 2012). As shown in Figure 2, the abundances of CHN+
compounds were 2.0%–3.6% in the haze days, which were much higher than those noted
in clean days (1.3%–1.4%), indicating higher contribution of CHN+ compounds to the
HULIS fraction during the haze episode. The $MW_w$ values for CHN+ compounds were
204–223, which were lower than those for the other groups (i.e., CHO+, CHON+) (Table
S5). However, the average $AI_{mod}$ values for N-bases (0.37–0.48) detected in the ESI+
mode were much higher than those for CHO+ (0.11–0.12) and CHON+ (0.20–0.22)
compounds, implying that these reduced CHN+ compounds exhibited more unsaturated
or aromatic structures.
To further understand the molecular distribution of CHN+ compounds during the
haze process, van Krevelen (VK) diagrams were constructed by plotting the H/C ratio
versus N/C ratio (Figure S2). It was obvious that this plot could separate the compound
classes with different degree of AI. As shown in Figure S2, compounds (denoted in black

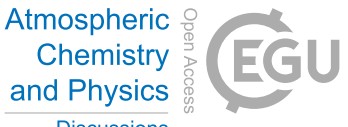

color) in the upper region of the VK diagram had one N atom with DBE = 0, indicating
that they are aliphatic amines. It can be noted from Table S7 that the aliphatic group
presented the lowest abundance in all the samples, suggesting that the CHN+ compounds
possessed comparatively lower aliphatic structures. Olefinic compounds showed the
highest abundance in the four samples, which accounted for 37%–51% of the total CHN+
compounds. Importantly, a large proportion of the compounds (>39%) exhibited high
degree of AI (AI > 0.5) (Figure S2 and Table S7), suggesting a large amounts of aromatic
structure and N-heterocyclic ring in HULIS. Moreover, the CHN+ compounds in haze-I
days presented obviously lower content of aromatic structures than those in haze-II,
clean-I, and clean-II days, signifying the relatively high contribution of fossil fuel
combustion (which generally emits more low-aromatic CHN compounds) during the haze
bloom episode(Song et al., 2022). In addition, the CHN+ group also constituted a large
proportion of BBOA (Table S6), which indicated the significant contribution of BB.
However, it must be noted that a relatively lower content of BBOA was detected during
haze-I days, which was consistent with the changing trends of CHON- or CHON+
compounds during the haze episode. These results suggested the relatively lower
contribution of BB during haze-I days, because quiet and stable weather conditions can
prevent regional transport of BB sources during this stage (Wu et al., 2018b).

**3.4. Factors influencing light absorption and molecular characteristics of HULIS during the haze bloom-decay process**

As described earlier, the light absorption properties of HULIS exhibited obvious
variation during the haze bloom-decay process. The average $Abs_{365}$ value for HULIS was
$0.55 \pm 0.06$ M m$^{-1}$ in clean-I days, which first increased to $3.4 \pm 1.5$ M m$^{-1}$ in haze-I days,



then slowly decreased to 2.6 $\pm$ 0.85 M m$^{-1}$ in haze-II days, and finally rapidly declined to
0.64 $\pm$ 0.32 M m$^{-1}$ in clean-II days. In general, the light absorption of HULIS can be
related to their chemical and molecular properties that are influenced by factors such as
sources, secondary formation, and aging process. The results of principal component
analysis (PCA) obviously showed a positive loading for principal component 1 (PC1),
and the Abs$_{365}$ values for HULIS were clustered with EC, K$_{bb}^{+}$, Lev, NH$_4^{+}$, and NO$_3^{-}$
(Figure 5). These results suggested that BB and other sources such as new particle
formation could contribute to light absorption of HULIS (Huang et al., 2014; An et al.,
2019; Song et al., 2019). Similarly, the findings of Pearson correlation coefficient
analysis revealed that the Abs$_{365}$ values for HULIS exhibited significant positive
correlations with K$_{bb}^{+}$ (r = 0.728, p < 0.01) and Lev (r = 0.800, p < 0.01) (Table S8). As
Lev and K$_{bb}^{+}$ are generally considered as tracers derived from BB, these results suggested
the significant contribution of BB to light absorption of HULIS. This observation was
also supported by the abundance of BBOA compounds detected in all the four HULIS
samples (Table S6). The significant positive relationships between the Abs$_{365}$ values for
HULIS and secondary ions (i.e., NO$_3^{-}$ (r = 0.702, p < 0.01), SO$_4^{2-}$ (r = 0.554, p < 0.05),
and NH$_4^{+}$ (r = 0.899, p < 0.01)) indicated the important impact of secondary formation on
the light absorption of HULIS. Besides, the Abs$_{365}$ values for HULIS were also strongly
correlated with NO$_2$, O$_3$, and NO$_2$, which confirmed the important impact of atmospheric
oxidation reactions on the light absorption of HULIS.

It must be noted that MAE$_{365}$ is a key parameter signifying the light absorption

ability of HULIS. As listed in Table S2, the MAE$_{365}$ values for HULIS varied in different
stages, and were lower in haze days owing to the variation in the chemical and molecular





composition of HULIS during the haze bloom-decay process. Furthermore, the $AI_{mod}$
values for HULIS varied in different stages (Tables S4 and S5), and were relatively lower
in haze days, indicating that haze HULIS have comparatively lower degree of
conjugation or aromaticity. This finding suggested that the HULIS compounds may
undergo higher oxidation and present longer aging period under specific quiet and stable
weather conditions during the haze episode, causing a decline in chromophores and
reduction in the light absorption capacity of HULIS (Lin et al., 2017). Besides, the
accumulated contribution of organic compounds from vehicle emission and secondary
chemical reactions of bio-VOCs may also dilute light-absorbing compounds in haze
HULIS (Tang et al., 2020; Liu et al., 2016).

Lin et al. (2018) reported that potential light-absorbing chromophores can be

determined in the region between $DBE = 0.5 \times C$ (linear conjugated polyenes $C_xH_yC_2$)
and $DBE = 0.9 \times C$ (fullerene-like hydrocarbons). In the present study, most of the high-
intensity CHON, CHO, and CHN compounds with high AI values were clustered in
potential BrC chromophore region (Figures S3 and S4), which mainly comprised CHON
(46%−50% in ESI- mode and 56%−62% in ESI+ mode, respectively) and CHO (44%−48%
in ESI- mode and 29%−38% in ESI+ mode, respectively) compounds (Table 1).
Although the contribution of CHN+ compounds to BrC was relatively lower, the content
of potential chromophores among the total CHN+ compounds was higher than those in
CHON+ and CHO+ compounds. Therefore, these three groups of light-absorbing
compounds (i.e., CHON+, CHN+, and CHO+ compounds) were further examined. As
shown in Table 1, the $Int_C/Int_{BrC}$ values of CHO− (i.e., content of CHO− chromophores
in the total chromophores) decreased from 48% to 44% whereas the $Int_C/Int_{BrC}$ values of



CHON− increased from 46% to 50% during the haze bloom process. These findings
indicated that more NOCs chromophores were formed during this stage in which higher
$NO_2$ concentration may be preferred for the formation of N-containing chromophores
such as nitrophenols. However, it must be noted that the proportions of both CHO− and
CHON− chromophores among the total identified compounds decreased from clean-I to
haze-I days, suggesting the occurrence of stronger photo-bleaching process during the
haze bloom stage (Zeng et al., 2020). Likewise, both CHO+ and CHON+ compounds
presented similar variation during the entire study period. In addition, the CHN+
compounds also exhibited higher $Int_C/Int_{BrC}$ values during the haze bloom process and
suggesting the accumulated contribution from localcombustion process. Furthermore, the
proportion of CHON+ chromophores in the total CHON+ compounds increased with the
decreasing content of CHN+ chromophores, may implying that some aromatic CHN
compounds were transformed to CHON+ compounds during the aging process.

**4. Conclusions**

This study investigated the evolution of light absorption and molecular properties of

HULIS during a winter haze bloom-decay process, and examined the key factors
affecting the light absorption of HULIS in Guangzhou, China. The results showed that
HULIS exhibited significant variation in light absorption during the haze bloom-decay
process. First, higher $Abs_{365}$ values were observed in haze days, indicating the presence
of significant amounts of light-absorbing organic compounds during the haze episode.
However, the $MAE_{365}$ values for HULIS in haze days were relatively lower than those in
clean days, suggesting the light absorption capabilities of HULIS were weakened during





the haze event. Furthermore, CHON and CHO compounds, exhibiting relatively higher
degree of conjugated structure, were the most abundant groups in all the HULIS samples,
and were also the major contributors to light absorption capacity of HULIS. Importantly,
the molecular properties of HULIS dynamically varied during the entire haze episode.
When compared with HULIS in clean days, those in haze days presented relatively lower
$AI_{mod}$ values and higher $O/C_w$, $O/N_w$, and $O/S_w$ ratios, suggesting the predominance of
compounds with low aromaticity and higher oxidation in HULIS during haze episode.
These results indicated that HULIS compounds undergo relatively stronger oxidation and
longer aging process during the haze process. Moreover, PCA and Pearson correlation
analysis revealed that BB and secondary chemical formation both contributed to the
variation in the light absorption properties of HULIS. Both primary sources (such as
accumulated contribution of organic compounds formed from local traffic emission) and
secondary sources (such as stronger chemical reactions) led to the rapid increase in
HULIS during the haze bloom days. However, longer periods for oxidation and aging of
HULIS compounds were observed during the haze episode, and some potential BrC
chromophores were degraded. In addition, the chemical reactions of bio-VOCs such as
isoprene also diluted the light-absorbing compounds in HULIS.

Thus, the present study provides novel insights into the light and molecular

evolution of HULIS during haze event, which are important for predicting the
environmental and climatic effects of HULIS. However, as this study examined only one
haze bloom-decay process in winter in Guangzhou, the results obtained may be not
adequate for understanding all the haze episodes in South China. Therefore, there is a





need for a comprehensive investigation of haze episode in different seasons and regions
in future.

**Data availability**

The    research    data    are    available    in    the    Harvard    Dataverse
(https://doi.org/10.7910/DVN/DYGYQT, Song, 2022).

**Author contributions.** J. Song and P. Peng designed the research together. C, Zou, T.
Cao, and M. Li carried out the $PM_{2.5}$ sampling experiments. C, Zou and T. Cao extracted
and analyzed the WSOC and HULIS samples. B. Jiang analyzed the HULIS samples by
FT-ICR MS. C. Zou and J. Song wrote the paper. J. Li, X. Ding, Z Yu, and G. Zhang
commented and revised the paper.

**Competing interests.** The authors declare that they have no conflict of interest

**Acknowledgments.** This study was supported by the National Natural Science
Foundation of China (42192514 and 41977188), Guangdong Foundation for Program of
Science and Technology Research (2020B1212060053), and Guangdong Foundation for
Program of Science and Technology Research (2019B121205006).

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


**Table 1.** Formular number of potential BrC chromophores and the intensity ratios of each group of
potential BrC in total potential BrC and each group of total identified formulas, respectively.

| Samples | Elemental compositions | Number | ESI- $Int_C/Int_{BrC}$ | $Int_{BrC,i}/Int_{bulk}$ | Elemental compositions | Number | ESI+ $Int_C/Int_{BrC}$ | $Int_{BrC,i}/Int_{bulk}$ |
|---|---|---|---|---|---|---|---|---|
| Clean-I | CHO- | 424 | **0.48** | **0.25** | CHO+ | 263 | **0.37** | 0.07 |
| | CHON- | 773 | **0.46** | **0.53** | CHON+ | 480 | **0.56** | **0.15** |
| | CHOS- | 63 | 0.03 | 0.05 | CHN+ | 79 | 0.07 | **0.56** |
| | CHONS- | 43 | 0.03 | 0.08 | all in ESI+ | 822 | | **0.11** |
| | all in ESI- | 1303 | | **0.26** | | | | |
| Haze-I | CHO- | 356 | **0.44** | **0.21** | CHO+ | 244 | **0.29** | 0.09 |
| | CHON- | 791 | **0.50** | **0.45** | CHON+ | 614 | **0.62** | **0.22** |
| | CHOS- | 43 | 0.03 | 0.03 | CHN+ | 94 | 0.09 | **0.39** |
| | CHONS- | 39 | 0.03 | 0.07 | all in ESI+ | 952 | | **0.16** |
| | all in ESI- | 1229 | | **0.22** | | | | |
| Haze-II | CHO- | 444 | **0.45** | **0.26** | CHO+ | 333 | **0.34** | 0.06 |
| | CHON- | 941 | **0.49** | **0.49** | CHON+ | 595 | **0.56** | **0.13** |
| | CHOS- | 67 | 0.03 | 0.03 | CHN+ | 89 | 0.1 | **0.48** |
| | CHONS- | 78 | 0.03 | 0.07 | all in ESI+ | 1017 | | **0.10** |
| | all in ESI- | 1530 | | **0.25** | | | | |
| Clean-II | CHO- | 391 | **0.46** | **0.27** | CHO+ | 234 | **0.38** | 0.09 |
| | CHON- | 707 | **0.48** | **0.59** | CHON+ | 462 | **0.56** | **0.18** |
| | CHOS- | 64 | 0.03 | 0.05 | CHN+ | 75 | 0.06 | **0.57** |
| | CHONS- | 49 | 0.03 | 0.10 | all in ESI+ | 771 | | **0.13** |
| | all in ESI- | 1211 | | **0.29** | | | | |

$Int_C$: the intensity of each group of identified potential BrC;
$Int_{BrC}$: the sum intensity of identified potential BrC;
$Int_{Bulk}$: the sum intensity of each group of total identified formulas.

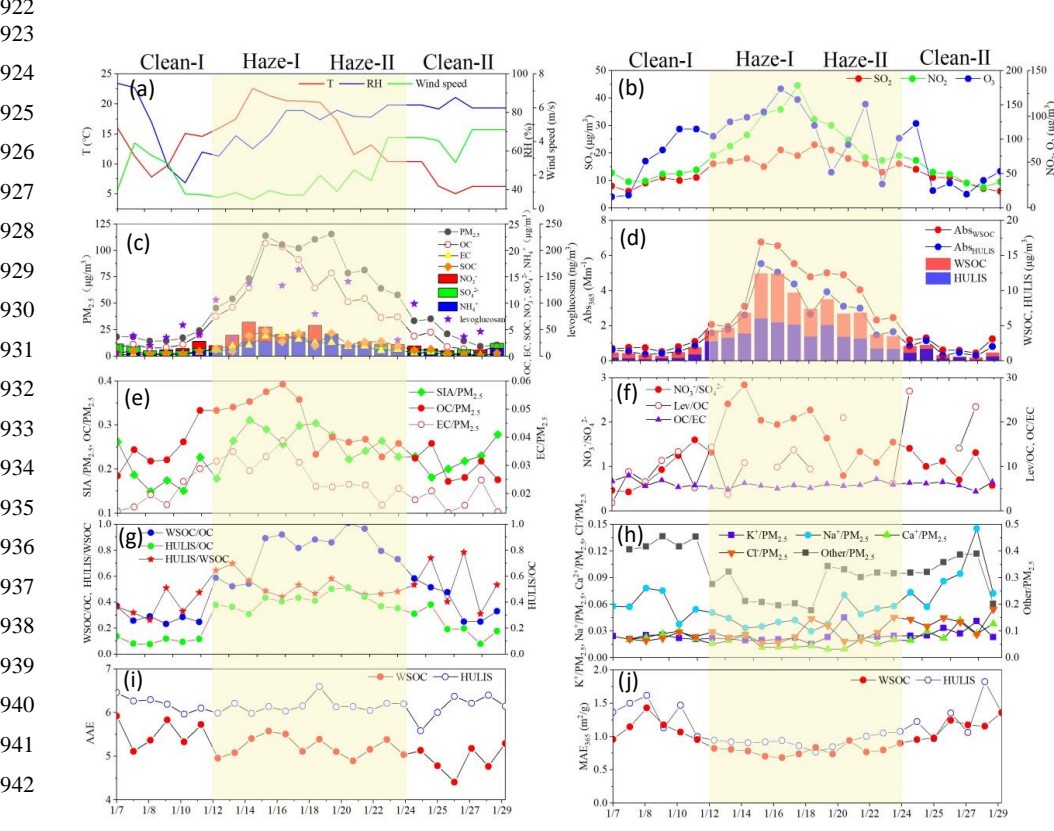

**Figure 1.** Temporal variation in meteorological parameters, concentrations of chemical composition, and optical properties (Abs$_{365}$, MAE$_{365}$, and AAE) of water-soluble BrC in the PM$_{2.5}$ samples.





951

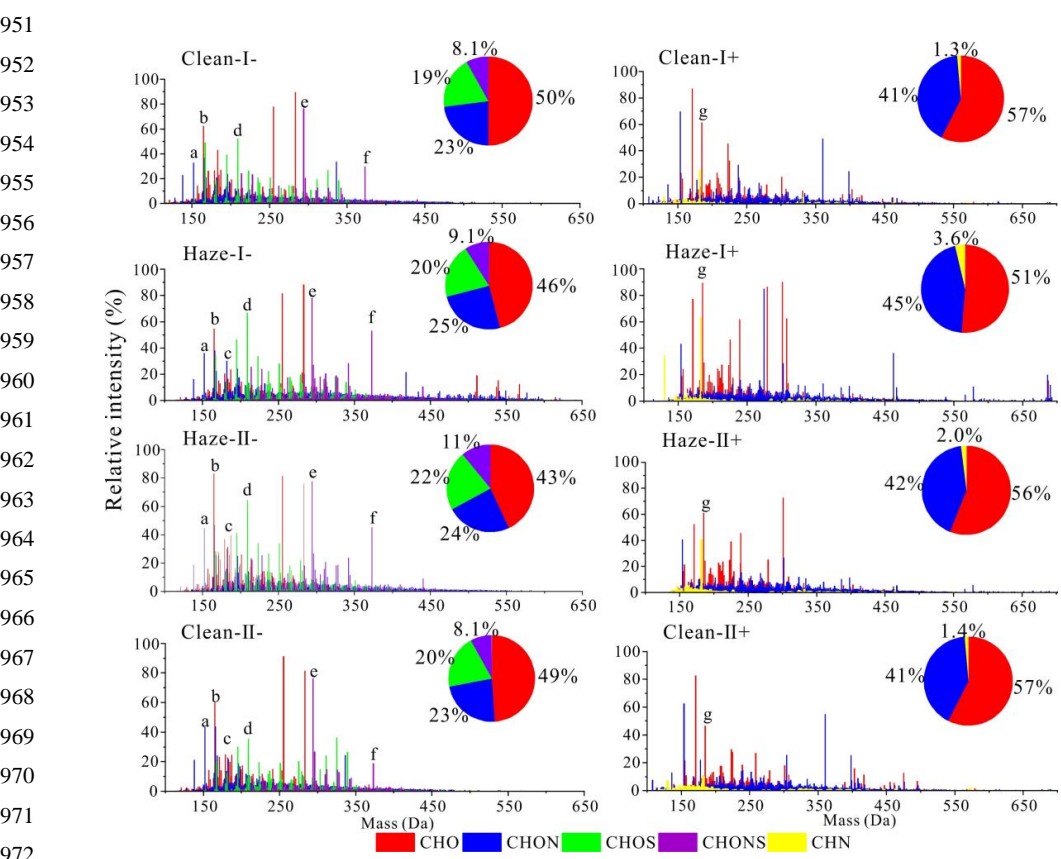

**Figure 2.** Mass spectra of HULIS detected in ESI- and ESI+ modes during the haze

process. The pie charts represent the intensity percent of different compound groups.



**Figure 3.** Carbon oxidation state (OSc) plots for CHO- and CHO+. Formulas with black,

green, blue, and red are assigned to aliphatic (AI = 0), olefinic (0< AI <0.5), aromatic

(0.5≤ AI <0.67), and condensed aromatic (AI ≥0.67) species (Koch and Dittmar, 2006),

respectively.





**Figure 4.** Carbon oxidation state (OSc) plots for CHON- and CHON+. Formulas with

black, green, blue, and red are assigned to aliphatic (AI = 0), olefinic (0< AI <0.5),

aromatic (0.5≤ AI <0.67), and condensed aromatic (AI ≥0.67) species (Koch and Dittmar,

2006), respective.





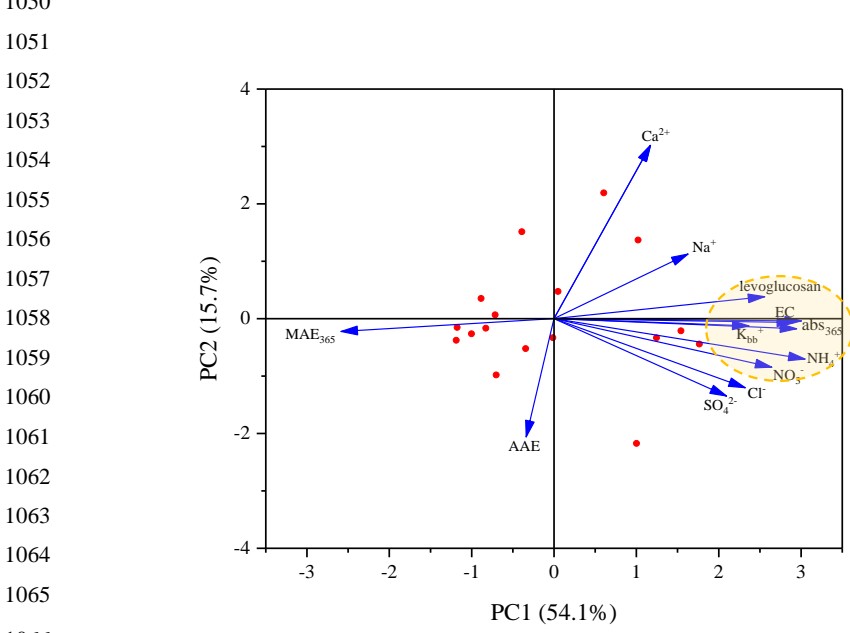

**Figure 5.** Principal component analysis results for the optical properties of HULIS and chemical compositions of $PM_{2.5}$.