# Peer review of "Measurement report: Changes in light absorption and molecular"

_Atmospheric Chemistry and Physics, 2022_

## Author Comment (AC1)

**Response to Anonymous Referee #1**

This manuscript describes measurements of light-absorption properties and chemical composition of ambient water-soluble HULIS samples collected during haze episodes and clean periods. The measurements involve comprehensive chemical analyses including carbon analysis (OC/EC and TOC), ion chromatography for inorganic ions, and ESI FTICR MS for organics. The light absorption properties were quantified using UV-vis spectrometry.

General comments:

- The data set produced in this study, especially the ESI FTICR MS data, is extensive and informative. However, there are various instances where assertions are made that are not supported by the data, and are at some points contradictory with other assertions in different parts of the manuscript. Please see specific examples under 'Specific comments' below.

Re: We appreciated the reviewer for the constructive and valuable comments, which is of great help to improve the quality of the manuscript. According to your 'Specific comments', we have carefully and thoughtfully revised the manuscript, and responded to all comments point by point, and explained how the reviewers' comments and suggestions were addressed in the current version of the manuscript.

- The term HULIS as used in this manuscript needs to be better defined. HULIS is a vague term – much like brown carbon – that has been used to refer to different things in different studies. Here, HULIS is obtained based on an extraction procedure that isolates the less polar fraction (~50%) of WSOC. It would be helpful for the reader to explicitly indicate in the methods section that this definition is operational, and also contrast the definition of HULIS in this study with other studies. This is important because the results are compared to multiple previous studies on HULIS, and it should be noted that not all HULIS are defined the same way.

Re: Thanks. We agreed with your comments that although the term "HULIS" has been used in many literatures, this concept is still vague, which may refer to different things in different studies. Therefore, the term HULIS should be defined in the manuscript, which are important when compared it with those in previous studies.

In this study, HULIS was isolated based on a water-extraction and SPE method that has been widely used by researchers in atmospheric science and environment (Lin et al., 2012; Fan et al., 2012, 2013; Zou et al., 2020; Jiang et al., 2020; Qin et al., 2022). According to your comments, we have added the operational definition of HULIS in the Method section. The revisions stated in the revised manuscript are as below:

"It is noted that the HULIS here is the hydrophobic portion of water-soluble organic matter, which can be isolated with different types of SPE columns (e.g., HLB, C-18, DEAE, XAD-8, and PPL) (Fan et al., 2012, 2013; Lin et al., 2012; Zou et al., 2020; Jiang et al., 2020; Qin et al., 2022). Although each resin type has its special chemical properties, the hydrophobic HULIS isolated with different sorbents were similar in chemical, molecular properties based on previous studies (Fan et al., 2012, 2013; Zou et al., 2020). Therefore, for better comparison with other studies, the hydrophobic fractions isolated by SPE methods were all termed as HULIS in the present paper." Please see Lines 151-158.

Reference:

Fan, X. J., Song, J. Z., and Peng, P. A.: Comparison of isolation and quantification methods to measure humic-like substances (HULIS) in atmospheric particles, Atmos. Environ., 60, 366–374, 10.1016/j.atmosenv.2012.06.063, 2012.

Fan, X., Song, J., Peng, P.: Comparative study for separation of atmospheric humiclike substance (HULIS) by ENVI-18, HLB, XAD-8 and DEAE sorbents: elemental composition, FT-IR, 1H-NMR and off-line thermochemolysis with tetramethylammonium hydroxide (TMAH). Chemosphere 93, 1710–1719, 10.1016/j.chemosphere.2013.05.045, 2013.

Jiang, H., Li, J., Chen, D., Tang, J., Cheng, Z., Mo, Y., Su, T., Tian, C., Jiang, B., Liao, Y., and Zhang, G.: Biomass burning organic aerosols significantly influence the light absorption properties of polarity-dependent organic compounds in the Pearl River Delta Region, China, Environ Int, 144, 106079, 10.1016/j.envint.2020.106079, 2020.

Lin, P., Rincon, A. G., Kalberer, M., and Yu, J. Z.: Elemental composition of HULIS in the Pearl River Delta Region, China: results inferred from positive and negative electrospray high resolution mass spectrometric data, Environ Sci Technol, 46, 7454-7462, 10.1021/es300285d, 2012.

Qin, J., Zhang, L., Qin, Y., Shi, S., Li, J., Gao, Y., Tan, J., and Wang, X.: pH-Dependent Chemical Transformations of Humic-Like Substances and Further Cognitions Revealed by Optical Methods. Environ Sci Technol, 56, 7578-7587, 10.1021/acs.est.1c07729, 2022.

Zou, C., Li, M., Cao, T., Zhu, M., Fan, X., Peng, S., Song, J., Jiang, B., Jia, W., Yu, C., Song, H., Yu, Z., Li, J., Zhang, G., and Peng, P. a.: Comparison of solid phase extraction methods for the measurement of humic-like substances (HULIS) in atmospheric particles, Atmos Environ, 225, 117370, 10.1016/j.atmosenv.2020.117370, 2020.

Specific comments:
- Section 2.4 and 2.5 should be combined: ESI-MS is also chemical analysis.

Re: Thanks. We have combined Section 2.4 and 2.5 as "2.4. Chemical analysis" in the current manuscript. Please see Line 165.

- Line 165: The manuscript presents results of $PM_{2.5}$ concentrations, but there is not description of how the $PM_{2.5}$ concentrations are measured in the methods section.

Re: Thanks for your comments. In this study, the $PM_{2.5}$ concentrations were determined by weighing the filters before and after collection. The details have been added in the Method section of the current manuscript. Please see Lines 125-130.

▪ Figure 1: There are inconsistencies in the x-axis values: the distance between the major ticks changes between 1 day (e.g. 1/24 – 1/25) and 2 days (e.g. 1/10 – 1/12).

Re: Sorry for this error. We have corrected that in the Figure 1.

▪ Figure 1f: how come the Lev/OC value are larger than 1? Lev is one of many OC species.

Re: Sorry, it is clerical error. We have revised "Lev/OC" in the right vertical axis to "Lev/OC ($\times 10^{-3}$)" in the Figure 1f.

▪ Line 218-219: The statement that Lev/OC increased in haze-II is not accurate. There are 2 data point for Lev/OC in haze-II (Figure 1f): one is higher than haze-I and one is lower than haze-I.

Re: We apologized for this error. Based on your comment, we have revised that in the current manuscript. Please see Lines 241-243.

▪ Line 232-233: This is not valid. AAE is a measure of the wavelength dependence of light absorption, not the magnitude of light absorption.

Re: Thanks for your comment. We agreed with that AAE is a measure of the wavelength dependence of light absorption, not the magnitude of light absorption. As shown in Figure 1i, the AAE values for HULIS were higher than those for WSOC in the same sample. We think that this difference may be related with the light-absorbing organic species in the isolated HULIS fractions have strong wavelength dependence than those in the original WSOC. We have revised that in the current manuscript. Please see Lines 254-256.

▪ Line 248-260: The statement on line 251 that MAE of HULIS is generally higher than WSOC is not valid. The values for HULIS (1.1 +/- 0.27) and WSOC (1 +/- 0.21) are virtually the same. In Lines 254-256, MAE of HULIS (1.1 +/- 0.27) is said to be "comparable" to other values ranging between 0.91 and 1.84. Then in line 257-259, MAE of 0.91 is said to be "much lower" than MAE of 1.3. These statements are subjective and inconsistent.

Re: Thanks for your comments. At first, we agreed with the comment that the $MAE_{365}$ values for HULIS (1.1 ± 0.27) and WSOC (1.0 ± 0.21) are virtually the same. Accordingly, we have revised this sentence to "'the average $MAE_{365}$ value for WSOC was $1.0 ± 0.21$ m$^2$ gC$^{-1}$ ($0.68–1.3$ m$^2$ gC$^{-1}$), nearly same to $1.1± 0.27$ m$^2$ gC$^{-1}$ ($0.77–1.8$ m$^2$ gC$^{-1}$) for HULIS, during the entire sampling period" in the current manuscript. Please see Lines 271-273.

In addition, for the statement in Lines 254-256, I want to say is that the $MAE_{365}$ values of HULIS (1.1 ± 0.27) in this study are dropped in the ranges between 0.91 and 1.84 reported in previous studies. The "comparable" is an inaccurate word, so we have revised that in the current manuscript. Please see Lines 274-275.

Finally, for the statement in Lines 257-259, we think it is right. As shown in Table S2, the $MAE_{365}$ values for HULIS were $0.91±0.03$ and $0.95 ± 0.11$ m$^2$ gC$^{-1}$ in haze-I and haze-II days, respectively, which were lower than those ($1.3 ± 0.22$ and $1.3 ± 0.27$ m$^2$ gC$^{-1}$, respectively) observed in clean-I and clean-II days.

▪ Line 263-266: The argument that stagnant conditions lead to prolonged oxidation thus lower MAE for haze versus clean days is not convincing. It is not clear that the PM sampled during the haze days had longer atmospheric lifetime / OH exposure. What if the PM in the clean days had more contribution from long-range transported PM?

Re: Thanks for your comments. We agreed with that the argument that stagnant conditions lead to prolonged oxidation thus lower MAE for haze versus clean days is not convincing because it is not clear if the PM sampled during the haze days or the clean days had longer atmospheric lifetime. According to the variation of meteorological parameters and atmospheric trace gases (e.g., ozone) and molecular properties of HULIS during the entire sampling period, we think that the relative lower $MAE_{365}$ values for haze HULIS may potentially contribute to the enhanced oxidation reaction that was derived by the increased ozone levels and high temperature and relative humidity during haze days (Figure 1). This stronger oxidation process would lead the chromophores containing C=C unsaturated bond to be severely degraded. Accordingly, we have revised that in the current manuscript. Please see Lines 284-288.

- Line 285-286: It is not clear how the presence of these 3 molecules suggests contribution from biomass burning and vehicular emissions.

Re: Thanks. In our study, these 3 molecules (i.e., Compounds a ($C_7H_7NO_3$) and b ($C_8H_6O_4$), and d($C_8H_{18}O_4S$) with stronger arbitrary abundance were identified, which may derived from biomass burning and vehicular emissions as reported in previous studies (Table S3) (Mohr et al., 2013; Riva et al., 2015; Blair et al., 2017). Therefore, the detection of these molecules in this study suggested some contribution from biomass burning and vehicular emissions. We have clarified that in the current manuscript. Please see Lines 303-306.

References:

Mohr, C., Lopez-Hilfiker, F. D., Zotter, P., Prevot, A. S., Xu, L., Ng, N. L., Herndon, S. C., Williams, L. R., Franklin, J. P., Zahniser, M. S., Worsnop, D. R., Knighton, W. B., Aiken, A. C., Gorkowski, K. J., Dubey, M. K., Allan, J. D., and Thornton, J. A.: Contribution of nitrated phenols to wood burning brown carbon light absorption in Detling, United Kingdom during winter time, Environ Sci Technol, 47, 6316-6324, 10.1021/es400683v, 2013.

Matthieu Riva, Ellis S. Robinson, Emilie Perraudin, Neil M. Donahue, and Eric Villenave. Photochemical Aging of Secondary Organic Aerosols Generated from the Photooxidation of Polycyclic Aromatic Hydrocarbons in the Gas-Phase[J]. Environmental Science & Technology: ES&T, 49, 5407-5416, 10.1021/acs.est.5b00442, 2015.

Blair S. L., Macmillan, A. C., Drozd G. T., Goldstein A. H., Chu R. K., Ljiljana P., Shaw J. B., Tolic Nikola, Lin Peng., Laskin J., Laskin A., and Sergey A. Nizkorodov. Molecular characterization of organosulfur compounds in biodiesel and diesel fuel secondary organic aerosol. American Chemical Society. Environ. Sci. Technol., 51, 119-127, 10.1021/acs.est.6b03304, 2017

▪ Line 308-318: This paragraph mentions that HULIS in haze days had higher MW than in clean days, and makes the point that high MW HULIS is more resistant to chemical transformation. This is in contrast with the assertions in section 3.2 that MAE on haze days were smaller than on clean days because HULIS on haze days underwent more chemical transformation.

Re: Thanks for the good comments. As mentioned in this study, HULIS in haze days had higher MW values than those in clean days, but the $MAE_{365}$ values on haze days were smaller than on clean days (Table S2, S4, and S5). We think these results are scientifically reasonable in the present study: (1) $MAE_{365}$ is a key parameter signifying the light absorption ability of HULIS or BrC. According to previous studies, the $MAE_{365}$ values were mainly affected by their unsaturated aromatic structures and they didn't exhibit significant relationship with the MW of HULIS or BrC (Song et al, 2019; Zeng et al., 2021; Jiang et al., 2021). As indicated in Table S4 and S5, although HULIS had higher MW during haze days, the $AI_{mod}$ values of haze HULIS were relatively lower. This result indicated that the haze HULIS were characterized by comparatively lower degree of conjugation or aromaticity, therefore, the $MAE_{365}$ values on haze days were smaller than on clean days is scientifically reasonable. (2) The HULIS is a class of highly complex organic compounds, which consists of various types of aromatic and aliphatic molecules. In general, the organic molecules

containing unsaturated bonds (e.g., aromatic structures, olefins) are relatively labile than those saturated aliphatic compounds (e.g., aliphatic acids), which are easy to be degraded during the atmospheric oxidation process (Claflin et al., 2018). Therefore, it is reasonable that the enhanced oxidation reaction during haze days degraded more aromatic structures and lead to relative lower $MAE_{365}$ values for haze HULIS. (3) Base on previous studies, the low MW compounds are more susceptible to atmospheric oxidation processes, while the high MW compounds have relatively higher chemical resistance (Di Lorenzo et al., 2016; Wong et al., 2017), therefore the enhanced oxidation reaction during haze days also lead to the enrichment of high MW HULIS compounds in haze days than in clean days. We have clarified that in the current manuscript. Please see Lines 335-342.

References:

Claflin, M. S.; Ziemann, P. J. Identification and quantitation of aerosol products of the reaction of β-pinene with $NO_3$ radicals and implications for gas- and particle-phase reaction mechanisms. J Phys Chem A 2018, 122 (14), 3640-3652.

Di Lorenzo, R. A.; Washenfelder, R. A., Attwood, A. R., Guo, H., Xu, L., Ng, N. L., Weber, R. J., Baumann, K., Edgerton, E., and Young, C. J.: Molecular-Size-Separated Brown Carbon Absorption for BiomassBurning Aerosol at Multiple Field Sites. Environ Sci Technol, 51, 3128−3137, 10.1021/acs.est.6b06160, 2017.

Jiang, H., Li, J., Sun, R., Tian, C., Tang, J., Jiang, B., Liao, Y., Chen, C. E., and Zhang, G.: Molecular Dynamics and Light Absorption Properties of Atmospheric Dissolved Organic Matter, Environ. Sci. Technol., 55, 10268–10279, 10.1021/acs.est.1c01770, 2021.

Song, J., Li, M., Fan, X., Zou, C., Zhu, M., Jiang, B., Yu, Z., Jia, W., Liao, Y., Peng, P.: Molecular Characterization of Water- and Methanol-Soluble Organic Compounds Emitted from Residential Coal Combustion Using Ultrahigh-Resolution Electrospray Ionization Fourier Transform Ion Cyclotron Resonance Mass Spectrometry. Environ Sci Technol, 53, 13607-13617, 10.1021/acs.est.9b04331, 2019.

Wong, J. P. S., Nenes, A., and Weber, R. J.: Changes in Light Absorptivity of Molecular Weight Separated Brown Carbon Due to Photolytic Aging, Environ Sci Technol, 51, 8414-8421, 10.1021/acs.est.7b01739, 2017.

Zeng, Y., Ning, Y., Shen, Z., Zhang, L., Zhang, T., Lei, Y., Zhang, Q., Li, G., Xu, H., Ho, S. S. H., Cao, J.: The roles of N, S, and O in molecular absorption features of brown carbon in $PM_{2.5}$ in a typical semi-arid megacity in Northwestern China. J Geophys Res Atmos, 126, 10.1029/2021JD034791, 2021.

▪ Line 319-330: This paragraph makes the point that lower AI_mod for haze days can be due to photooxidation during haze days and explain that lower MAE for haze days. How is this assertion reconciled with the larger MW and resistance to oxidation mentioned in the previous paragraph?

Re: Thanks for your comment. As discussed above, we think it is scientific reasonable. As indicated in this study, the molecular properties of HULIS in different stages of haze process exhibited some observable differences. The HULIS samples in haze days presented relatively higher MW and relatively lower $AI_{mod,w}$ values than those in clean days (Table S4). These results indicated that the haze HULIS have relatively higher molecular weight, but their aromaticity were lower than the clean HULIS. We think that these results could be related to the evolution of different types of HULIS molecules during atmospheric oxidation process. On the one hand, the organic molecules containing unsaturated aromatic structures are relatively labile than those saturated aliphatic molecules, which are easy to be degraded during the atmospheric oxidation process (Claflin et al., 2018). Therefore, the enhanced oxidation during haze days would degrade more aromatic structures and lead to relative lower $AI_{mod,w}$ values for haze HULIS. On the other hand, the low molecular weight compounds are more susceptible to atmospheric oxidation processes, while the high MW organic compounds have relatively higher chemical resistance (Di Lorenzo et al., 2016; Wong et al., 2017), therefore the enhanced oxidation reaction during haze days also lead to the haze HULIS are characterized with relative higher MW values. Therefore, the

haze HULIS have relative lower $AI_{mod}$ and higher MW values than the clean HULIS is scientific reasonable.

References:

Claflin, M. S.; Ziemann, P. J. Identification and quantitation of aerosol products of the reaction of β-pinene with $NO_3$ radicals and implications for gas- and particle-phase reaction mechanisms. J Phys Chem A 2018, 122 (14), 3640-3652.

Wong, J. P. S., Tsagkaraki, M., Tsiodra, I., Mihalopoulos, N., Violaki, K., Kanakidou, M., Sciare, J., Nenes, A., and Weber, R. J.: Atmospheric evolution of molecular-weight-separated brown carbon from biomass burning, Atmos Chem Phys, 19, 7319–7334, 10.5194/acp-19-7319-2019, 2019.

- Line 374-378: It does not look like the statement "relatively low BBOA content" is supported by the data in Figure 3. Most of the molecules are clustered in the region identified as BBOA. In any case, previous parts of the manuscript mention BBOA as being an important contributor to HULIS measured in this study, but this paragraph mentions that traffic sources are more important.

Re: We apologized for this error. At first, we agreed with your comment that "relatively low BBOA content" is an inaccurate description. Accordingly, we have removed that in the current manuscript.

In addition, we also agreed with your comment "BBOA is an important contributor to HULIS as measured in this study", however some HULIS species derived from the traffic sources were also identified. As shown in Table S6 and Figure 3, relative contents of traffic sources in haze days are higher than in clean days, but BBOA is still the most important contributor to HULIS in the present study. We have revised that in the current manuscript. Please see Lines 389-391, 397-400.

Minor comments:

- Line 80-83: The statement talks about 'recent years' but is supported by a reference from 2014. A newer reference is needed.

Re: Thanks. We have added three new references in the current manuscript. Please see Lines 86-87.

References:

An, Z., Huang, R. J., Zhang, R., Tie, X., Li, G., Cao, J., Zhou, W., Shi, Z., Han, Y., Gu, Z., and Ji, Y.: Severe haze in northern China: A synergy of anthropogenic emissions and atmospheric processes, Proc Natl Acad Sci USA, 116, 8657-8666, 10.1073/pnas.1900125116, 2019.

Li, K., Jacob, D. J., Liao, H., Shen, L., Zhang, Q., and Bates, K.H.: Anthropogenic drivers of 2013–2017 trends in summer surface ozone in China, P. Natl. Acad. Sci. USA, 116, 422–427, https://doi.org/10.1073/pnas.1812168116, 2019.

Yang, X., Lu, K., Ma, X., Gao, Y., Tan, Z., Wang, H., Chen, X., Li, X., Huang, X., He, L., Tang, M., Zhu, B., Chen, S., Dong, H., Zeng, L., and Zhang, Y.: Radical chemistry in the Pearl River Delta: observations and modeling of OH and $HO_2$ radicals in Shenzhen in 2018. Atmos. Chem. Phys., 22, 12525-12542, 10.5194/acp-22-12525-2022, 2022

- Line 91: What is meant by 'exact'?

Re: We are sorry for this inaccurate word. We have deleted it in the current manuscript. Please see Line 101.

- Line 99-102: This sentence is not comprehensible.

Re: Thanks. We have rewritten this sentence in the current manuscript. Please see Lines 109-112.

- Line 180-182: This statement is not valid. Wind speed alone does not dictate stability (See stability classifications by Turner 1970). In fact, for an unstable atmosphere, increasing wind speed makes the atmosphere less unstable.

Re: Thanks for your comments. We have deleted it in the current manuscript.

- Line 277-279: Vague statement. In what sense are the peaks "comparable" with peaks from other studies?

Re: Sorry for this vague statement. We have deleted that in the current manuscript.

- Line 337-339: I assume you mean biomass burning aerosol (not biomass burning mixture). In any case, what does "comparable" mean here?

Re: Yes. It is biomass burning aerosol. In addition, the "comparable" is an inaccurate word. We have rewritten that in the current manuscript. Please see Lines 361-363.

---

## Author Comment (AC2)

**Response to Anonymous Referee #2**

This manuscript reports the measurements of the water-soluble organic carbon (WSOC) and WS-HULIS fraction of $PM_{2.5}$ samples collected in a developed region with dense populations during a haze event; the authors conducted a comprehensive analysis of the chemical composition and light absorption with their samples. They investigated the evolution of light absorption and molecular properties of these samples during one haze cycle (clean-haze bloom-haze decay-clean). While I think the subject is very interesting, there are several issues with this manuscript, including the analysis and conclusions, which are detailed below.

Re: Thanks for the constructive and valuable comments, which is of great help to improve the quality of the manuscript. According to your comments, we have carefully and thoughtfully revised the manuscript, and responded to all comments point by point, and explained how the reviewers' comments and suggestions are handled in the current manuscript.

**General comments:**

▪    It will be necessary for the author to provide more details in the results and discussion, especially when making deductions and conclusions. This manuscript provides a very comprehensive dataset of the chemical and optical analysis of their samples. However, in the data interpretation, some of their conclusions/statements are given too simply and vaguely, which needs to be supported with more details and be more specific. For example, in explaining the variance in properties (MW, MAE, etc) of HULIS and WSOC obtained during different stages of the haze event (e.g., haze day vs clean days), the authors mostly attribute these differences to statements such as "effects of aging/oxidation/degradation" without further explanations or details. Since the aging process involves many different pathways and mechanisms, the authors will need to be more specific in the results when explaining the data other than just simply stating "aging".

Re: Thanks for your comments. We agreed to your comments that some of statements are given too simply and vaguely in the data interpretation, especially for the variance in properties (MW, MAE, etc) of HULIS and WSOC obtained during different stages of the haze event. In the current manuscript, we have revised that and provided more specific interpretations in the results and discussion. Please see Lines 284-288, 335-342, 361-363, 543-545, 592-593, 598.

- The "HULIS fraction" used in this manuscript needs to be better described and defined since this is the major substance studied here. Adding some brief descriptions in the introduction would be necessary. HULIS was first reported as macromolecular organic substances in atmospheric aerosol particles; and when used to refer to the light absorbing properties of the atmospheric aerosols, "HULIS" – humic-like substance, is more describing the similarities in the light absorption of the light absorbing organic carbons (e.g., brown carbon) with humic substances, which is a light absorption that sharply decreases from UV to visible wavelength. Also, many different methods for the extraction/isolation of HULIS have been reported, and the potential effects of the extraction/isolation procedure on the chemical/physical nature of HULIS have aroused concerns as well. For instance, the HULIS part could be large molecules formed by intermolecular force (aggregates) and the procedure (extraction solvent, PH adjustment, SPE, etc.) would largely change it. I think a discussion of the nature of HULIS and a brief explanation of your choice for the isolation method, at least under the circumstances of this manuscript, is important for the data interpretation later in the result section.

Re: Good comments. We agreed with your comments that a discussion of the nature of HULIS and a brief explanation of the isolation method should be added in the manuscript. This is important for the data interpretation later in the result section and compared it with those in previous studies. According to your comments, we have added some descriptions in the Introduction and Method sections. The revisions stated

in our revised manuscript are as below:

"Water-soluble humic-like substances (HULIS), belonging to a class of highly complex organic compounds with physical/chemical properties similar to humic substances in natural environments" and "They are thought to be comprised of aromatic structures containing aliphatic side side chains and oxygenated functional groups such as hydroxyl, carboxyl, nitrate, and organosulfate groups". Please see Lines 42-44, 47-50.

"It is noted that the HULIS here is the hydrophobic portion of water-soluble organic matter, which can be isolated with different types of SPE columns (e.g., HLB, C-18, DEAE, XAD-8, and PPL) (Fan et al., 2012, 2013; Lin et al., 2012; Zou et al., 2020; Jiang et al., 2020; Qin et al., 2022). Although each resin type has its special chemical properties, the hydrophobic HULIS isolated with different sorbents were similar in chemical, molecular properties based on previous studies (Fan et al., 2012, 2013; Zou et al., 2020). Therefore, for better comparison with other studies, the hydrophobic fractions isolated by SPE methods were all termed as HULIS in the present paper." Please see Lines 151-158.

Reference:

Fan, X. J., Song, J. Z., and Peng, P. A.: Comparison of isolation and quantification methods to measure humic-like substances (HULIS) in atmospheric particles, Atmos. Environ., 60, 366–374, 10.1016/j.atmosenv.2012.06.063, 2012.

Fan, X., Song, J., Peng, P.: Comparative study for separation of atmospheric humiclike substance (HULIS) by ENVI-18, HLB, XAD-8 and DEAE sorbents: elemental composition, FT-IR, 1H-NMR and off-line thermochemolysis with tetramethylammonium hydroxide (TMAH). Chemosphere 93, 1710–1719, 10.1016/j.chemosphere.2013.05.045, 2013.

Jiang, H., Li, J., Chen, D., Tang, J., Cheng, Z., Mo, Y., Su, T., Tian, C., Jiang, B., Liao, Y., and Zhang, G.: Biomass burning organic aerosols significantly influence the light absorption

properties of polarity-dependent organic compounds in the Pearl River Delta Region, China, Environ Int, 144, 106079, 10.1016/j.envint.2020.106079, 2020.

Lin, P., Rincon, A. G., Kalberer, M., and Yu, J. Z.: Elemental composition of HULIS in the Pearl River Delta Region, China: results inferred from positive and negative electrospray high resolution mass spectrometric data, Environ Sci Technol, 46, 7454-7462, 10.1021/es300285d, 2012.

Qin, J., Zhang, L., Qin, Y.,  Shi, S., Li, J., Gao,  Y., Tan,  J., and Wang, X.: pH-Dependent Chemical Transformations of Humic-Like Substances and Further Cognitions Revealed by Optical Methods. Environ Sci Technol, 56, 7578-7587, 10.1021/acs.est.1c07729, 2022.

Zou, C., Li, M., Cao, T., Zhu, M., Fan, X., Peng, S., Song, J., Jiang, B., Jia, W., Yu, C., Song, H., Yu, Z., Li, J., Zhang, G., and Peng, P. a.: Comparison of solid phase extraction methods for the measurement of humic-like substances (HULIS) in atmospheric particles, Atmos Environ, 225, 117370, 10.1016/j.atmosenv.2020.117370, 2020.

- In order to highlight the novelty of this manuscript, it's better to have a brief summary of the work about HULIS fractions/WSOC of $PM_{2.5}$ in the Pearl River Delta with emphasis on the major contributions of this manuscript to the current state of knowledge regarding this topic. I do notice there is one sentence mentioning that there are previous works regarding this topic (line 83-86), however, I think brief descriptions of the referred studies are necessary. Also, is this the only work studying the chemical/optical evolution of WSOC/HULIS/Brown carbon collected in PRD? If not, what have the other studies done?

Re: Thanks. According to your comments, we have added brief descriptions of the referred studies. In addition, we also added new references related with the chemical/optical evolution of WSOC/HULIS/Brown carbon collected in PRD in the current manuscript. The detailed revisions are:

"Several studies have investigated the optical, chemical, and molecular properties of HULIS in the PRD region (Lin et al., 2010, 2012; Fan et al., 2016; Liu et al., 2018;

Jiang et al., 2020, 2021a,b). For example, the studies on the temporal variations of water-soluble HULIS in Guangzhou indicated that HULIS had higher concentrations and mass absorption efficiencies (MAE365) in the winter, which were attributed to the increasing contribution of BB and secondary nitrate formation in the winter monsoon period (Fan et al., 2016; Jiang et al., 2020, 2021a). In addition, the molecular composition of HULIS (and BrC) in the PRD region were also investigated and demonstrated that the levels of unsaturated and aromatic structures are the important factor influencing their light absorption properties (Jiang et al., 2020, 2021b)." Please see Lines 87-96.

References:

Fan, X., Song, J., and Peng, P.: Temporal variations of the abundance and optical properties of water soluble Humic-Like Substances (HULIS) in PM$_{2.5}$ at Guangzhou, China, Atmos Res, 172-173, 8-15, doi:10.1016/j.atmosres.2015.12.024, 2016.

Jiang, H., Li, J., Chen, D., Tang, J., Cheng, Z., Mo, Y., Su, T., Tian, C., Jiang, B., Liao, Y., and Zhang, G.: Biomass burning organic aerosols significantly influence the light absorption properties of polarity-dependent organic compounds in the Pearl River Delta Region, China, Environ Int, 144, 106079, 10.1016/j.envint.2020.106079, 2020.

Jiang, H., Li, J., Sun, R., Liu, G., Tian, C., et al. Determining the sources and transport of brown carbon using radionuclide tracers and modeling, J Geophys Res Atmos, 126, e2021JD034616, doi:org/10.1029/2021JD034616, 2021a.

Jiang, H., Li, J., Sun, R., Tian, C., Tang, J., Jiang, B., Liao, Y., Chen, C. E., and Zhang, G.: Molecular dynamics and light absorption properties of atmospheric dissolved organic matter, Environ Sci Technol, 55, 10268–10279, https://doi.org/10.1021/acs.est.1c01770, 2021b.

Lin, P.; Engling, G.; Yu, J. Z.: Humic-like substances in fresh emissions of rice straw burning and in ambient aerosols in the Pearl River Delta Region, China. Atmos Chem Phys, 10, 6487−6500, 10.5194/acp-10-6487-2010, 2010.

Lin, P., Rincon, A. G., Kalberer, M., and Yu, J. Z.: Elemental composition of HULIS in the Pearl River Delta Region, China: results inferred from positive and negative electrospray high

resolution mass spectrometric data, Environ Sci Technol, 46, 7454-7462, 10.1021/es300285d, 2012.

Liu, J., Mo, Y., Ding, P., Li, J., Shen, C., and Zhang, G.: Dual carbon isotopes ((14)C and (13)C) and optical properties of WSOC and HULIS-C during winter in Guangzhou, China, Sci Total Environ, 633, 1571-1578, 10.1016/j.scitotenv.2018.03.293, 2018.

**Specific comments:**

Line 37: comment for writing — "stronger" than what?

Re: This is inaccurate word. We have deleted it in the current manuscript.

Line 44: "natural environment" It is better to be more specific here about what environment (e.g., natural aquatic / soil environment).

Re: Thanks. We revised it to "natural aquatic/soil environment" in the current manuscript. Please see Line 44.

Line 45-46: "> 70% of light absorption in water-soluble brown carbon (BrC)" This need to be more specific about the wavelength/wavelength range or what parameters they used to compare (e.g., mass absorption coefficients, etc), if possible.

Re: Thanks. In this study, "> 70% of light absorption in water-soluble brown carbon (BrC)" was calculated by light absorption at 365 nm. We have clarified it in the current manuscript. Please see Line 46.

Line 114: "Field blank samples were collected without power on." Does this mean that the blank filter is "conditioned" in the air sample holder rather than conditioned in the sampling environment (e.g., passing particle-free air through the filter)? If so, can you explain why you chose this way as "blank control"?

Re: Yes. In this study, the blank filter is "conditioned" in the air sample holder on the sampling site. This method was chosen because: (1) In this study, field blank samples were collected in the air sample holder and then were analysed exactly as the procedure for the $PM_{2.5}$ samples. This is a good method for estimating the potential pollution during the $PM_{2.5}$ sampling operation and the lab's operation (including filter sample weighting, water extraction, SPE isolation, etc.) and has been recommended by US EPA (Watson et al., 1998) and widely used for correcting the filter $PM_{2.5}$ samples in many studies (Li et al., 2020; Zhu et al., 2020; Jiang et al., 2021; Deng et al., 2022; Zhan et al., 2022). (2) in this study, the $PM_{2.5}$ sampler is a high-volume sampler ($1.0 \ m^3 \ min^{-1}$), therefore, it is impracticable to collect blank filter using a particle-free air during 24-h field sampling period. Therefore, "conditioned in the air sample holder" was used as "blank control" in this study.

References:

Deng, J. J., Ma, H., Wang, X. F., Zhong, S. J., Zhang, Z. M., Zhu, J. L., Fan, Y. B., Hu, W., Wu, L. B., Li, X. D., Ren, L. J., Pavuluri, C. M., Pan, X. L., Sun, Y. L., Wang, Z. F., Kawamura, K., and Fu, P. Q.: Measurement report: Optical properties and sources of water-soluble brown carbon in Tianjin, North China insights from organic molecular compositions, Atmos Chem Phys, 22, 6449-6470, 10.5194/acp-22-6449-2022, 2022.

Jiang, H., Li, J., Sun, R., Liu, G.,Tian, C., Tang, J., Cheng Z., Zhu S., Guangcai Zhong G., Ding X., and Zhang G.. Determining the sources and transport of brown carbon using radionuclide tracers and modeling. Journal of Geophysical Research: Atmospheres, 126, e2021JD034616. https://doi.
    org/10.1029/2021JD034616, 2021.

Li, J. J., Zhang, Q., Wang, G. H., Li, J., Wu, C., Liu, L., Wang, J. Y., Jiang, W. Q., Li, L. J., Ho, K. F., and Cao, J. J.: Optical properties and molecular compositions of water-soluble and water-insoluble brown carbon (BrC) aerosols in northwest China, Atmos Chem Phys, 20, 4889-4904, 10.5194/acp-20-4889-2020, 2020.

Watson, J.G., J.C. Chow, H. Moosmüller, M. Green, N. Frank, and M. Pitchford: Guidance for Using Continuous Monitors in PM2.5 Monitoring Networks: Draft 03/06/98. Prepared for Office of Air Quality Planning and Standards, U.S. Environmental Protection Agency, Research Triangle Park, N.C., 27711 by Desert Research Institute, Reno, Nev, 1998.

Zhan, Y. A., Li, J. L., Tsona, N. T., Chen, B., Yan, C. Q., George, C., and Du, L.: Seasonal variation of water-soluble brown carbon in Qingdao, China: Impacts from marine and terrestrial emissions, Environ Res, 10.1016/j.envres.2022.113144, 212, 2022.

Zhu, C. S., Li, L. J., Huang, H., Dai, W. T., Lei, Y. L., Qu, Y., Huang, R. J., Wang, Q. Y., Shen, Z. X., and Cao, J. J.: n-Alkanes and PAHs in the Southeastern Tibetan Plateau: Characteristics and Correlations With Brown Carbon Light Absorption, J Geophys Res-Atmos, 125, 10.1029/2020JD032666, 2020.

Line 121-122: "Briefly, portions of the $PM_{2.5}$ samples ($100 \ cm^2$) were ultrasonically extracted with 50 mL of ultrapure water for 30 min. The extracts were filtered through a 0.22-μm PTFE syringe filter and then adjusted to pH of 2 with HCl…". I didn't find the description of the WSOC fraction here. Or is the extraction of WSOC the same as described for HULIS except for the PH adjustment and following procedure? I think this needs to be clarified here.

Re: Thanks. In this study, portions of the $PM_{2.5}$ samples ($100 \ cm^2$) were ultrasonically extracted with 50 mL of ultrapure water for 30 min. The extracts were filtered through a 0.22-μm PTFE syringe filter to remove the suspended insoluble particles. About 50 mL of water extracts were obtained from each sample, of which 20 mL was used for the isolation and analysis of HULIS, 20 mL for analysis of water soluble organic carbon (WSOC), and the remainder for the analysis of inorganic ions, respectively. Then the 20 mL water extracts were adjusted to 2 with HCl and introduced into a pre-conditioned HLB cartridge. The hydrophilic fraction (i.e., inorganic ions, high-polar organic acids, etc) was removed with ultrapure water, whereas the relatively hydrophobic HULIS fraction was retained and eluted with 2% (v/v) ammonia/methanol. Finally, HULIS solution was evaporated to dryness with a gentle

N$_2$ stream and redissolved with ultrapure water for the analysis. We have clarified it in the current manuscript. Please see Lines 138-144.

Line 232-233 "This difference may be related to the higher enrichment of light-absorbing organic species in HULIS." The statement is oversimplified and needs more specific explanation and evidence to describe why the enrichment of light-absorbing OC doesn't simply enhance the light absorption at certain wavelengths but the wavelength dependence (AAE).

Re: Thanks for your comments. In this study, the AAE values for HULIS were higher than those for WSOC in the same sample. We think that this difference may be related with the light-absorbing organic species in the isolated HULIS fractions have strong wavelength dependence than those in the original WSOC. We have revised that in the current manuscript. Please see Lines 254-256.

Line 263-265: it might need to be more specific about the "secondary oxidation reaction" and "photolytic aging" here. It is "prolonged" or "enhanced" oxidation during haze? for example, the increased ozone levels associated with high levels of PM$_{2.5}$ could be a drive for oxidation during haze days.

Re: Thanks for your comments. We agreed with that it need to be more specific about the "secondary oxidation reaction" and "photolytic aging" here. Although it is not clear if the PM sampled during the haze days or the clean days had longer atmospheric lifetime, we think the "enhanced" oxidation during haze could lead to strong aging of HULIS, which was mainly derived by the increased ozone levels and high temperature and relative humidity during the haze days. Accordingly, we have revised that in the current manuscript. Please see Lines 284-288.

Line 315-318: The author needs to mention that the samples used in Dasari and Wong's work are from different sources (mostly biomass burning aerosols) and are

different from the samples used in this manuscript. In addition, the method for molecular weight estimation used in Wong's paper is also very different from this method (size-exclusion chromatography), in which the MW is estimated by the SEC column retention time, and it's also highly dependent on the column, the mobile phase, and the sample itself (e.g., polarity, aggregation, etc). Also, line 317-318 is a bit self-contradictory: HULIS in haze undergo stronger oxidation, which usually leads to fast degradation. However, "a longer aging process" means higher stability (longer lifetime), which means the HULIS is more stable during haze days. Or, is the "longer aging" simply implying that because they have a higher MW, they will have a longer lifetime in the atmosphere? There needs some clarification.

Re: Thanks for your comments. We agreed with that the samples used in the references are different from the samples used in this manuscript and the method for molecular weight estimation is also different the method in our study. We have clarified that in the current manuscript. Please see Lines 335-339.

For Lines 317-318: We are sorry for this vague interpretation. In this study, the MW values for HULIS during haze days were higher than those in clean days, as indicated in Table S4 and S5. These differences may be related to the enhanced oxidation reaction of HULIS during the haze days. In general, HULIS is class of highly complex organic compounds, with MW ranges from dozens to thousands. According to previous studies, the low MW fractions are more susceptible to bleaching and high MW fractions are recalcitrant under atmospheric oxidation processes of biomass burning aerosols (Di Lorenzo, et al., 2017; Wong et al., 2017; Dasari et al., 2019). Therefore, the enhanced oxidation reaction during haze days would lead to the enrichment of high MW compounds in HULIS during haze days. We have clarified that in the current manuscript. Please see Lines 335-342.

References:

Dasari, S., Andersson, A., Bikkina, S., Holmstrand, H., Budhavant, K., Satheesh, S., Asmi, E., Kesti, J., Backman, J., Salam, A., Bisht, D. S., Tiwari, S., Hameed, Z., and Gustafsson, Ö.: Photochemical degradation affects the light absorption of water-soluble brown carbon in the South Asian outflow, Sci Adv, 5, eaau8066, doi: 10.1126/sciadv.aau8066, 2019.

Di Lorenzo, R. A.; Washenfelder, R. A., Attwood, A. R., Guo, H., Xu, L., Ng, N. L., Weber, R. J., Baumann, K., Edgerton, E., and Young, C. J.: Molecular-Size-Separated Brown Carbon Absorption for BiomassBurning Aerosol at Multiple Field Sites. Environ Sci Technol, 51, 3128−3137, 10.1021/acs.est.6b06160, 2017.

Wong, J. P. S., Nenes, A., and Weber, R. J.: Changes in Light Absorptivity of Molecular Weight Separated Brown Carbon Due to Photolytic Aging, Environ Sci Technol, 51, 8414-8421, 10.1021/acs.est.7b01739, 2017.

Even if the author wanted to use these two references to support their inference about the higher stability of HULIS in this manuscript, they didn't try to explain the variance in HULIS MW from different stages of the haze event (haze days vs clean days).

Re: Thanks for your comments. We have explained the variance in HULIS MW from different stages of the haze event in the current manuscript. Please see Lines 335-342.

Line 324-325: "These differences can be attributed to bleaching or degradation of aromatic compounds." As I understand it, "degradation" is one pathway that leads to bleaching, i.e., the degradation/oxidation of aromatic compounds could result in the bleaching of HULIS. There might need some clarification or rephrasing.

Re: Thanks for your comments. We have revised that in the current manuscript. Please see Lines 348-350.

Line 573-574: It is unclear if the "longer aging process" here means "a longer lifetime." Is this an inference or an observation?

Re: Thanks for your comments. The "longer aging process" is an inaccurate statement. In this study, the HULIS compounds should undergo relatively stronger oxidation during the haze days. We have revised that in the current manuscript. Please see Lines 592-593.

---

## Author Response (AR2)

**Comments:**

I appreciate the author's responses and their efforts to address the questions in the revised manuscript. However, I regret to say that I still think some of the discussions in the result part are not clear enough. I have some extra questions as listed below:

Re: Thanks for your valuable comments, which is of great help to improve the quality of the manuscript. According to your comments, we have carefully revised the manuscript, and responded to all comments point by point, and explained how the reviewers' comments and suggestions are handled in the current manuscript.

General comments:

1. In section 2.1, line 128-129, samples were collected on the quartz fiber filters and the $PM_{2.5}$ concentrations were determined by weighing the filters before and after collection. According to past experience, quartz fiber filters are very fragile and flaky; the loss of fibers from the filter during the handling and sampling will be inevitable and thus make this type of filter VERY challenging for mass weighing, and probably impossible for the accuracy of 0.01 mg claimed here. Therefore, it is not clear to me how the accuracy of the $PM_{2.5}$ concentrations can be guaranteed in this study.

Re: Thanks. We agreed with your comments that "quartz fiber filters are very fragile and flaky; the loss of fibers from the filter during the handling and sampling will be inevitable and thus make this type of filter VERY challenging for mass weighing, and probably impossible for the accuracy of 0.01 mg claimed here". Based on your comments, we conducted a group of experiments to test the losing of fibers during the sample collection. In our experiment, the sample filters were put in the air sample holder for 5 min without pumping air, then the filters were wrapped in prebaked aluminum foil. Before and after sampling, the filters were weighed at 25°C and 50% RH on a microbalance. This treatment was repeated for three times. As indicated in

the following Table A, the weights of two PM$_{2.5}$ samples were 41.2±0.5 mg and 34.7±0.7 mg, respectively. No significant losing of fibers were observed.

The 0.01 mg is the Readability of the microbalance in this study, which is not the accuracy of the weighting of filter samples. We are sorry for this error in the original manuscript. In the current manuscript, we have revised that to "The mass accuracy achieved was < 2% based on triplicate analyses of filter samples". Please see Lines 128-129.

Table A: The weights of PM$_{2.5}$ filter samples determined at three times.

|  | Sample 1 (mg) | Sample 2 (mg) |
|---|---|---|
| Original sample | 41.3 | 34.3 |
| 1 | 41.7 | 35.2 |
| 2 | 41.2 | 35.4 |
| 3 | 40.5 | 34.0 |
| Average | 41.2 | 34.7 |
| Standard derivation (SD) | 0.5 | 0.7 |
| Relative standard derivation (RSD) | 1.2 | 2.0 |

2. Related to the first comment, the claimed accuracy is 0.01mg, which is 10 µg; however, the reported masses of PM$_{2.5}$ are data like 9-35 µg, 18+-3.3 µg, etc. This seems a bit strange to me.

Re: Thank for your question. In this study, the mass of PM$_{2.5}$ were shown as PM$_{2.5}$ concentration in air (µg m$^{-3}$). They were calculated based on the following equation:

$$C = \frac{W \times 1000}{V \times t}$$

Where, C is the concentration of PM$_{2.5}$ in air (µg m$^{-3}$), $W$ is the weight of PM$_{2.5}$ sample (mg), $V$ is the flow rate of PM$_{2.5}$ sampler (1.0 m$^3$ min$^{-1}$), $t$ is the sampling time (min). In this study, the weight of PM$_{2.5}$ samples collected were 13−166 mg, the calculated PM$_{2.5}$ concentrations were 9.3−115 µg m$^{-3}$. In the manuscript, the data such as "9-35, 18+-3.3", etc, is the PM$_{2.5}$ concentration (µg m$^{-3}$) rather than the PM$_{2.5}$ mass weight (mg). Therefore, it is scientifically reasonable.

3. Line 254-255: This added sentence is just iterating sentences 251-253 in a different format (i.e., a higher AAE value means a stronger wavelength dependence). To me, this is not explaining (i.e., higher wavelength dependence could result from higher molecular weights, since the HULIS fraction could isolate most of the large molecules from the WSOC; or it could also mean a higher degree of conjugation in molecular structure in HULIS, as they are the hydrophobic fraction, etc). in the first round of review, the previous suggestion of a more specific explanation of the results does not mean simply explaining the meaning of the factor itself.

Re: Thanks. We are apologized for our insufficient revision in the manuscript. In this study, AAE is a measure of the wavelength dependence of BrC light absorption, which appear to be related to the chemical composition of chromophores and the fitting wavelength ranges (Zhang et al., 2013; Chen et al., 2016; Park et al., 2018). Since the light absorption of BrC mostly occurs in near-UV and visible wavelengths, AAE (330-400 nm) was usually reported in many previous studies and also used in this study. As shown in Figure 1i, the AAE values for HULIS were obviously higher than those for WSOC in the same sample, indicating that light absorption of HULIS is more wavelength-dependent than that of WSOC. Similar results were also observed in previous studies (Park et al., 2018; Jiang et al., 2020; Cao et al., 2021). We think that this difference could be associated with the differences in chemical composition of chromophores in WSOC and HULIS. As shown in Table S2, the $E_{250}/E_{365}$ values of HULIS (5.3–5.6) all higher than that (4.4–5.1) of WSOC, suggested that the light-absorbing species in HULIS may have relative lower aromaticity and/or lower molecular weight than those in WSOC (Chen et al., 2016; Li and Hur, 2017). This difference may be attributed to a fraction of higher MW species remained in the HLB column due to irreversible adsorption and/or incomplete elution (Fan et al., 2012). As the results, the light-absorbing organic species in the HULIS fractions have relative higher absorption at UV and short visible wavelengths and relative lower absorption at long visible wavelengths, which resulting in relative higher AAE values. We have

revised that in the current manuscript. Please see Lines 249-260.

**Table S2.** Average values of $Abs_{365}$, $MAE_{365}$, AAE, and $E_{250}/E_{365}$ of WSOC and HULIS in $PM_{2.5}$ samples.

| | | Overall | Clean-I | Haze-I | Haze-II | Clean-II |
|---|---|---|---|---|---|---|
| $Abs_{365}$ | WSOC | 2.5 ± 2.0 | 0.76 ± 0.18 | 4.3 ± 2.0 | 3.9 ± 1.1 | 0.89 ± 0.35 |
| | HULIS | 1.8 ± 1.6 | 0.55 ± 0.06 | 3.4 ± 1.5 | 2.6 ± 0.85 | 0.64 ± 0.32 |
| $MAE_{365}$ | WSOC | 1.0 ± 0.21 | 1.1 ± 0.16 | 0.76 ± 0.05 | 0.83 ± 0.07 | 1.1 ± 0.14 |
| | HULIS | 1.1 ± 0.27 | 1.3 ± 0.22 | 0.91 ± 0.03 | 0.95 ± 0.11 | 1.3 ± 0.27 |
| AAE | WSOC | 5.2 ± 0.34 | 4.9 ± 0.55 | 5.0 ± 0.63 | 4.4 ± 0.20 | 4.7 ± 0.33 |
| | HULIS | 6.2 ± 0.20 | 6.2 ± 0.16 | 6.1 ± 0.09 | 6.2 ± 0.18 | 6.1 ± 0.28 |
| $E_{250}/E_{365}$ | WSOC | 4.8 ± 0.49 | 5.1 ± 0.70 | 4.9 ± 0.43 | 4.4 ± 0.15 | 4.8 ± 0.31 |
| | HULIS | 5.4 ± 0.35 | 5.6 ± 0.36 | 5.3 ± 0.15 | 5.4 ± 0.29 | 5.4 ± 0.52 |

References:

Cao, T., Li, M., Zou, C., Fan, X., Song, J., Jia, W., Yu, C., Yu, Z., Peng, P.: Chemical composition, optical properties, and oxidative potential of water-and methanol-soluble organic compounds emitted from the combustion of biomass materials and coal, Atmos Chem Phys, 21, 13187-13205, doi: 10.5194/acp-21-13187-2021, 2021.

Chen, Q., Ikemori, F., Mochida, M.: Light absorption and excitation-emission fluorescence of urban organic aerosol components and their relationship to chemical structure, Environ. Sci. Technol., 50, 10859-10868, doi: 10.1021/acs.est.6b02541, 2016.

Fan, X. J., Song, J. Z., and Peng, P. A.: Comparison of isolation and quantification methods to measure humic-like substances (HULIS) in atmospheric particles, Atmos. Environ., 60, 366–374, 10.1016/j.atmosenv.2012.06.063, 2012.

Jiang, H., Li, J., Chen, D., Tang, J., Cheng, Z., Mo, Y., Su, T., Tian, C., Jiang, B., Liao, Y., and Zhang, G. Biomass burning organic aerosols significantly influence the light absorption properties of polarity-dependent organic compounds in the Pearl River Delta Region, China. Environ. Int., 144, 106079, doi: 10.1016/j.envint.2020.106079, 2020.

Li, P. and Hur, J., Utilization of UV-Vis spectroscopy and related data analyses for

dissolved organic matter (DOM) studies: A review, Crit. Rev. Environ. Sci. Technol., 47, 131–154, doi: 10.1080/10643389.2017.1309186, 2017.

Park, S., Yua, G.-H., and Lee, S., Optical absorption characteristics of brown carbon aerosols during the KORUS-AQ campaign at an urban site, Atmos. Res., 203, 16–27, doi: 10.1016/j.atmosres.2017.12.002, 2018.

Zhang, X., Lin, Y.-H., Surratt, J.D., Weber, R.J., Sources, composition and absorption Ångstrom exponent of light-absorbing organic components in aerosol extracts from the Los Angeles Basin, Environ. Sci. Technol., 47, 3685-3693, doi: 10.1021/es305047b, 2013.

4. I found it interesting that the authors saw the AAE of HULIS fractions from all four stages are similar (line 256-257), whereas later in the manuscript, the analyses showed that HULIS collected from haze days have a slightly higher molecular weight and also higher oxidation of aromatics than those in clean days. Potentially, these two could lead to different trends in AAE: higher MW could lead to the shifting to a more humic-like structure (as stated in cited paper as Wong et al and Di Lorenzo et al), which might lead to a higher AAE; whereas the oxidation of aromatics could reduce the degree of conjugation, thus a less wavelength dependence. It would be better for the readers if the authors make this clearer.

Re: Thanks for your good comments. In this study, the AAE values of HULIS fractions from all four stages are similar, which could be related with the evolution of HULIS chromophores in different stages (Huang et al., 2018; Jiang et al., 2020; Deng et al., 2022). At first, the enhanced oxidation of aromatic species in haze days could lead to the bleaching or degradation of BrC chromophores, thus a less wavelength dependence (Forrister et al., 2015;Zhan et al., 2022). In contrast, the outburst of secondary organic aerosols and the photolysis of organic aerosols in haze days both tended to have higher AAE values (Saleh et al., 2013; Dasari et al., 2019). Consequently, the different trends in AAE were counterbalanced during the haze days, which resulting in no significant AAE variations were observed for HULIS fractions

from the four stages. This is also consistent with the trends of the $E_{250}/E_{365}$ ratios of HULIS in the four stages (Table S2). We have revised that in the current manuscript. Please see Lines 261-271.

References:

Deng, J., Ma, H., Wang, X., Zhong, S., Zhang, Z., Zhu, J., Fan, Y., Hu, W., Wu, L., Li, X., Ren, L., Pavuluri, C.M., Pan, X., Sun, Y., Wang, Z., Kawamura, K., and Fu, P.: Measurement report: Optical properties and sources of water-soluble brown carbon in Tianjin, North China –insights from organic molecular compositions, Atmos. Chem. Phys., 22, 6449–6470, doi: 10.5194/acp-22-6449-2022, 2022.

Dasari, S., Andersson, A., Bikkina, S., Holmstrand, H., Budhavant, K., Satheesh, S., Asmi, E., Kesti, J., Backman, J., Salam, A., Bisht, D. S., Tiwari, S., Hameed, Z., and Gustafsson, Ö.: Photochemical degradation affects the light absorption of water- soluble brown carbon in the South Asian outflow, Sci Adv, 5, eaau8066, doi: 10.1126/sciadv.aau8066, 2019.

Forrister, H., Liu, J., Scheuer, E., Dibb, J., Ziemba, L., Thornhill, K. L., Anderson, B., Diskin, G., Perring, A. E., Schwarz, P., Campuzano-Jost, P., Day, D. A., Palm, B. B., Jimenez, J. L., Nenes, A., and Weber, R. J.: Evolution of brown carbon in wildfire plumes, Geophys. Res. Lett., 42, 4623–4630, doi: 10.1002/2015gl063897, 2015.

Huang, R. J., Yang, L., Cao, J., Chen, Y., Chen, Q., Chen, Q., Li, Y. J., Duan, J., Zhu, C. C., Dai, W. T., Wang, K., Lin, C. S.;,Ni, H. Y., Corbin, J. C., Wu, Y. F., Zhang, R. J. Tie, X. X., Hoffmann, T., O'Dowd, C., and Dusek, U.: Brown carbon aerosol in urban Xi'an, Northwest China: The composition and light absorption properties, Environ Sci Technol, 52, 6825-6833, doi: 10.1021/acs.est.8b02386, 2018

Jiang, H., Li, J., Chen, D., Tang, J., Cheng, Z., Mo, Y., Su, T., Tian, C., Jiang, B., Liao, Y., and Zhang, G. Biomass burning organic aerosols significantly influence the light absorption properties of polarity-dependent organic compounds in the Pearl

River Delta Region, China. Environ. Int., 144, 106079, doi: 10.1016/j.envint.2020.106079, 2020.

Saleh, R., Hennigan, C. J., McMeeking, G. R., Chuang, W. K., Robinson, E. S., Coe, H., Donahue, N. M., and Robinson, A. L.: Absorptivity of brown carbon in fresh and photo-chemically aged biomass-burning emissions, Atmos. Chem. Phys., 13, 7683–7693, doi: 10.5194/acp-13-7683-2013, 2013.

Zhan, Y., Li, J., Tsona, N.T., Chen, B., Yan, C., George, C., and Du, L.: Seasonal variation of water-soluble brown carbon in Qingdao, China: Impacts from marine and terrestrial emissions, Environ. Res., 212, 113144, doi: 10.1016/j.envres.2022.113144, 2022.

5. The author might need to be careful in comparing the molecular weight (MW) with previous studies. For the size exclusion chromatography used in the cited papers, the analytical range of the column is usually from 250 Da ~ 75K Da. Considering the large range and the uncertainty associated with SEC working principle, (i.e., the MW is estimated from the log MW using a calibration curve), most of the MW reported (< 300 Da) here should be considered as "small fractions". Furthermore, in some cited papers (e.g., Di Lorenzo et al), the larger fraction of BrC is defined as MW larger than 500 Da, which means the analyzed fractions here are definitely "small" components.

Re: Thanks. We agreed with your comments that it should be very careful in comparing of the molecular weight (MW) with previous studies (e.g., Di Lorenzo et al, Wong et al., etc). In this study, the MW values of HULIS were determined with ESI FT-ICR MS. However, the MW measurements used in those papers are very different, in which the MW values were estimated by the SEC column retention time, and it's highly dependent on the column, the mobile phase, and the sample itself (e.g., polarity, aggregation, etc). Obviously, the theory of MW determination and the ranges of MW values are very different. Therefore, it is unreasonable for comparing the MW of HULIS with those previous studies (i.e., SEC method). Accordingly, we have removed that in the current manuscript. Please see Lines 347-348.

---

## Author Response (AR3)

**Response to Comments**

L 255: Change "As shown in Table S2, the E250/E365 values of HULIS (5.3−5.6) are higher than that (4.4−5.1) of WSOC, suggesting that the light-absorbing species in HULIS may have relatively lower aromaticity and/or lower molecular weight than those in WSOC … Therefore, the HULIS fractions exhibit relatively higher absorption at UV and short visible wavelengths and relatively lower absorption at long visible wavelengths, which results in relatively higher AAE values."

Re: Changed. Please refer to Lines 255-260 in the present manuscript.

L261: "…which could be related to the evolution of HULIS chromophores at different stages…"

Re: Changed. Please refer to Lines 261-262 in the present manuscript.

L267: change "…tended to the …" to "resulted in the …"

Re: Changed. Please refer to Line 267 in the present manuscript.

L269: change "which resulted in no significant AAE variations for HULIS in the entire sampling process."

Re: Changed. Please refer to Lines 269-270 in the present manuscript.